# iTRAQ-Based Proteomic Analysis of APP Transgenic Mouse Urine Exosomes

**DOI:** 10.3390/ijms24010672

**Published:** 2022-12-30

**Authors:** Xiaojing Zhou, Abdullah Md. Sheikh, Ken-ichi Matsumoto, Shingo Mitaki, Abu Zaffar Shibly, Yuchi Zhang, Garu A, Shozo Yano, Atsushi Nagai

**Affiliations:** 1Department of Neurology, Faculty of Medicine, Shimane University, 89-1 Enya-cho, Izumo 693-8501, Japan; 2Department of Laboratory Medicine, Faculty of Medicine, Shimane University, 89-1 Enya-cho, Izumo 693-8501, Japan; 3Department of Biosignaling and Radioisotope Experiment, Interdisciplinary Center for Science Research, Organization for Research, Shimane University, 89-1 Enya-cho, Izumo 693-8501, Japan

**Keywords:** Alzheimer’s disease, urine exosomes, amyloid beta, iTRAQ, proteomic analysis

## Abstract

Alzheimer’s disease (AD) is a common dementia disease in the elderly. To get a better understanding of the pathophysiology, we performed a proteomic analysis of the urine exosomes (U-exo) in AD model mice (J20). The polymer precipitation method was used to isolate U-exo from the urine of 3-month-old J20 and wild-type (WT) mice. Neuron-derived exosome (N-exo) was isolated from U-exo by immunoprecipitation. iTRAQ-based MALDI TOF MS/MS was used for proteomic analysis. The results showed that compared to WT, the levels of 61 and 92 proteins were increased in the J20 U-exo and N-exo, respectively. Gene ontology enrichment analysis demonstrated that the sphingolipid catabolic process, ceramide catabolic process, membrane lipid catabolic process, Aβ clearance, and Aβ metabolic process were highly enriched in U-exo and N-exo. Among these, Asah1 was shown to be the key protein in lipid metabolism, and clusterin, A*poE*, neprilysin, and ACE were related to Aβ metabolism and clearance. Furthermore, protein–protein interaction analysis identified four protein complexes where clusterin and A*poE* participated as partner proteins. Thus, J20 U-exo and N-exo contain proteins related to lipid- and Aβ-metabolism in the early stages of AD, providing a new insight into the underlying pathological mechanism of early AD.

## 1. Introduction

Alzheimer’s disease (AD) is a common dementia disease in the elderly population. The presence of extracellular amyloid plaques and intraneuronal neurofibrillary tangles in certain brain areas are considered the main pathological hallmarks of the disease [1]. The amyloid plaques mainly contain aggregated amyloid β (Aβ) peptide. Many genetic and animal studies have demonstrated the importance of Aβ in AD pathology [2]. In addition to amyloid plaques and neurofibrillary tangles, several other features including neurodegeneration, neuroinflammation, abnormal lipid metabolisms, and abnormal angiogenesis are seen in AD brains [3,4,5]. All of these processes are shown to have a link with Aβ deposition. For example, the Aβ peptide, mainly its aggregated form, has neurodegenerative and neuroinflammatory properties [6]. This peptide also influences the metabolisms of brain lipids including glycerophospholipids and sphingolipids. These changes in lipid metabolisms in turn affect their production and processing, causing a positive feedback cycle that increases the Aβ burden in the brain [7,8]. Additionally, such changes in lipid compositions are implicated in alteration of cell functions [9], such as cell–cell communications and signaling [10], metabolic functions [11], and protein quality control [12].

Increased production or decreased clearance is considered the cause of Aβ deposition. Increased production of the peptide is seen in the familial type of AD, where gene mutations alter Aβ-producing enzyme activities [13]. Additionally, mutations in amyloid β precursor protein (APP) increase the substrate, resulting in higher enzyme activity [14]. After production, the peptide is removed from the brain by enzymatic degradations and cell- and vessel-mediated clearance [15]. In sporadic AD, vessel-dependent Aβ clearance, where apolipoprotein E (A*poE)* has a crucial role, is suggested to be disturbed [16]. Hence, the peptide should be deposited mainly around the vessels [17]. However, intracellular aggregated Aβ can also be found in a widespread area of brain regions in AD [18,19]. How this aggregated peptide is deposited intracellularly and affects such wide areas of the brain is still under investigation. It is speculated that Aβ is transferred from cell to cell by certain mechanisms. In this respect, exosome-mediated transport could be the reason for such widespread distribution of Aβ. Indeed, recent reports showed the importance of exosome-mediated transport of Aβ in the pathology of AD [20,21].

Exosomes are extracellular vesicles with a diameter of 30–150 nm, which carry cargo-like signaling molecules, RNAs, and proteins. They are released from various types of cells into the extracellular space and transport the cargo to the target cells to modulate their functions. In addition to extracellular space, exosomes can be found in other body fluids, including blood, cerebrospinal fluid (CSF), peritoneal fluid, and excreta-like urine [22,23]. Cells of the central nervous system (CNS) including neurons, astrocytes, and microglia are reported to use exosomes to control the functions of other cell types [24,25] and participate in processes such as neurogenesis [26], neuroinflammation [27], angiogenesis, and neuroprotection [28]. To modulate such processes, exosome contents need to be altered depending on the disease pathology. Importantly, they are reported to cross the blood–brain barrier and can be isolated from peripheral blood [29,30,31,32]. Hence, studying neuron-derived exosomes from peripheral sources could be simpler and provide valuable information about the disease pathology.

The diagnosis of a disease is based on identifying the specific pathological changes, which are reflected in the specific biochemical and other changes in body fluids, excreta, or tissues. AD is a disease of the brain, and no biochemical changes are seen in body fluids or excreta, making diagnosis an important issue [33,34]. AD is largely diagnosed clinically since there are no commonly available laboratory tests, and tissue biopsies are not recommended for the disease [35]. A definitive diagnosis is done by postmortem analysis of the brain [36]. Recently, positron emission tomography (PET)-based diagnosis has been developed, where Aβ or tau ligands are used as probes. All of these PET-based diagnosis systems have limitations regarding availability, cost, and maintenance [37]. Analysis of CSF is also important for AD diagnosis. However, the main drawback is its invasiveness [38]. Since exosomes have the ability to cross the blood–brain barrier that reflects the pathological profile of the disease, it is conceivable that brain-derived exosomes isolated from peripheral fluid could be a valuable target as a diagnostic marker and used as a tool for disease diagnosis. Additionally, AD subjects showed impaired kidney functions [39,40,41]. As a result, the contents of urine exosomes might be changed, and neuron-derived exosomes could find a way to appear in the urine. Hence, we hypothesized that proteomic analysis of urine exosomes, specifically the neuron-derived exosomes, could provide valuable information about AD pathology and suitable targets for disease diagnosis.

To test this hypothesis, we isolated exosomes from the urine of an AD model mouse. Then, neuron-derived exosomes were separated from urine exosomes, and comprehensive proteomic and bioinformatic analyses were performed. We finally screened out acid ceramidase (Asah1) as a key protein in lipid metabolism and four other proteins specific to Aβ pathology, including clusterin, apolipoprotein E, neprilysin (Mme), and angiotensin converting enzyme (ACE), which contributes to uncovering the pathophysiology of AD and provides a new insight into the study of early diagnosis of AD.

## 2. Results

### 2.1. Characterization of Exosomes Isolated from Urine

The exosomes were isolated from urine (U-exo). Then, neuron-derived exosomes (N-exo) were purified from U-exo by immunoprecipitation using an anti-L1CAM antibody. Isolated exosomes were characterized using exosome markers including Tsg101, ALIX, CD9, and CD63. The Western blotting (WB) results demonstrated that both U-exo and N-exo isolated from WT and J20 mice were positive for Tsg101, ALIX, CD9, and CD63 (Figure 1A). Transmission electron microscopy (TEM) of the exosomes showed a typical exosomal spherical bilayer membrane structure in both U-exo and N-exo (Figure 1B). Additionally, the diameters of the U-exo and N-exo were also in the exosome diameter range of 30–150 nm. Among them, the average diameters of the U-exo and N-exo isolated from WT were 60.6 and 63.5 nm, respectively. In the case of J20, the average diameters were 67.4 nm and 70.4 nm in U-exo and N-exo, respectively.

### 2.2. Analysis of the Exosome Proteins by Mass Spectrometry

To identify and quantify the exosome proteins, matrix-assisted laser desorption ionization time-of-flight tandem mass spectrometry (MALDI TOF MS/MS) was performed. A total of 659 proteins with intensity values were detected in U-exo (Appendix A), and a total of 481 proteins with intensity values were detected in N-exo (Appendix A). In order to minimize the false positive identification of proteins, we selected the proteins with high confidence for subsequent analysis. For U-exo, 79 proteins were screened as identified proteins based on the following criteria: (1) FDR < 5%. (2) The number of distinct peptides with at least 95% confidence ≥ 1. (3) The protein can be detected with ratio at least two times in three biological replicates (Appendix A). In the case of N-exo, 117 proteins were screened as identified proteins based on the following criteria: (1) FDR < 5%. (2) The number of distinct peptides with at least 95% confidence ≥ 1. (3) The proteins can be detected with ratio (Appendix A). As most biochemical methods are prone to include technical variance, we considered an additional cutoff of a 1.2-fold change in the iTRAQ ratio (J20/WT) to select for different proteins. At last, 61 and 92 proteins were more abundant in the U-exo and N-exo of J20 mice compared to wild type mice, respectively (Table 1 and Table 2). Furthermore, after comparing the different proteins between U-exo and N-exo, we found that 41 proteins were common in both U-exo and N-exo, while 20 and 51 proteins are exclusively in U-exo and N-exo, respectively (Figure 2).

### 2.3. Analysis of Protein Subcellular Localization and Brain Expression

To characterize the exosomal proteins, an analysis of the protein subcellular localization was done. The results showed that 47% of the proteins of U-exo and 48% of the proteins of N-exo were extracellular proteins (Figure 3). Subcellular localization of other proteins was also similar between U-exo and N-exo (plasma membrane: U-exo- 34%, N-exo- 33%; cytosol: U-exo- 7%, N-exo- 9%; nucleus: U-exo-4%, N-exo-1%; mitochondria: U-exo-3%, N-exo-4%; lysosome: U-exo-1%, N-exo-1%) (Figure 3). Moreover, in order to evaluate the efficacy of the L1CAM antibody to separate N-exo from U-exo, we checked the brain expression of the identified proteins by web tool GenAtlas. Our results showed that 30% of U-exo proteins (24 proteins) can be expressed in the brain while 35% of N-exo proteins (41 proteins) can be expressed in the brain (Appendix A, Figure 4). Therefore, our results confirmed that N-exo contains more proteins that can be expressed in the brain. Furthermore, to further confirm the neuron origin of N-exo, we checked the neuron markers through the online database CellMarker 2.0. The results showed that there were 15 neuron markers in N-exo. Among them, seven neuron markers were shared in U-exo and N-exo, and another eight neuron markers were uniquely in N-exo (Table 3), suggesting that urine contains exosomes of neuron origin, and it can be further enriched in N-exo.

### 2.4. Gene Ontology (GO) Enrichment Analysis of the Common Different Exosome Proteins

In order to investigate the functions of U-exo and N-exo different proteins, the common different exosome proteins in both U-exo and N-exo were used for GO enrichment analysis and characterized in terms of biological process (BP), cellular component (CC), and molecular functions (MF). The results showed that 13 BP terms were enriched. Among them, nine BP terms were related with proteins, including two proteolysis terms (negative regulation of proteolysis and regulation of proteolysis), five enzyme activity terms (negative regulation of peptidase activity, negative regulation of hydrolase activity, regulation of peptidase activity, negative regulation of catalytic activity, and negative regulation of endopeptidase activity), one term protein maturation term, and one protein processing term. In addition, there were two BP terms related with circulatory system (vasculature development and regulation of blood pressure), one term with regulation of the renal system process and one term with negative regulation of secretion (Figure 5, Appendix A). Importantly, only 18 common different proteins were enriched in the BP terms, including pregnancy zone protein (Pzp), protein AMBP (Ambp), cathepsin B (Ctsb), growth arrest-specific protein 6 (Gas6), urokinase-type plasminogen activator (Plau), alpha-1-antitrypsin 1-4 (Serpina1d), alpha-1-antitrypsin 1-5 (Serpina1e), serine protease inhibitor A3K (Serpina3k), A*poE*, clusterin, neprilysin, ACE, aminopeptidase N (Anpep), kallikrein-1 (Klk1), meprin A subunit alpha (Mep1a), Actin (Actg1), complement factor D (Cfd), and major urinary protein 1 (Mup1) (Appendix A).

For the category cellular component, a total of 17 CC terms were enriched. Among them, 10 CC terms represented the characteristics of exosomes, including three membrane structure terms (brush border, brush border membrane, and cluster of actin-based cell projections) and seven extracellular domain terms (extracellular exosome, extracellular vesicle, extracellular organelle, extracellular membrane-bounded organelle, extracellular matrix, external encapsulating structure, and collagen-containing extracellular matrix). The other three CC terms (vacuole, lytic vacuole, and lysosome) suggested that our exosomes play a role in proteolysis. The remaining four CC terms (high-density lipoprotein particle, plasma lipoprotein particle, lipoprotein particle, and protein-lipid complex) implied that the two exosomes are related to lipid metabolism (Figure 5, Appendix A). In the case of molecular functions, a total of 17 MF terms were enriched. Among them, 13 enzyme activity MF terms were enriched, including the top three terms with smallest *p* value: endopeptidase activity, peptidase activity, and peptidase regulator activity. The other four MF terms reflected the binding functions, including zinc ion binding, peptide binding, amide binding, and pheromone binding (Figure 5, Appendix A).

### 2.5. Gene Ontology (GO) Enrichment Analysis of Identified Exosome Proteins

To unravel the functions of the identified proteins in U-exo and N-exo, GO enrichment analysis was performed. We considered highly enriched terms with a small *p*-value for analysis. In the case of U-exo, the results showed that 11 clusters of biological processes were highly enriched in identified proteins. Considering the *p* values and enrichment factor, the top two BP clusters were sphingolipid catabolic process and Aβ clearance. The clusters peptide catabolic process and wound healing also had high enrichment factors, but the *p* values were higher than the terms sphingolipid catabolic process and Aβ clearance. Regarding cellular components, eight CC clusters were highly enriched in U-exo proteins. Among them, terms including lytic vacuole, brush border membrane, and extracellular exosome were highly enriched with low *p* values. Terms including high-density lipoprotein and phagocytic vacuole were highly enriched, but the *p* values were higher than the lytic vacuole or brush border membrane. Regarding molecular functions, the enzyme activity terms including peptidase activity, hydrolase activity, and serine-type exopeptidase activity were highly enriched with low *p* value. The terms including immunoglobulin binding, low-density lipoprotein particle receptor binding, and aspartic-type endopeptidase activity were also enriched. However, the *p* values were not as low as peptidase activity (Figure 6, Appendix A).

Next, the proteins of N-exo were analyzed. In the case of biological process, 20 BP clusters were highly enriched in identified N-exo proteins. Considering the *p* values and enrichment factors, the top two BP terms were the peptide catabolic process and sphingolipid catabolic process. The *p* values of the term carbohydrate metabolic process and inflammatory response were high, but enrichment factors were not as high as the peptide catabolic process. The terms peptide lipid complex, glycoside catabolic process, and Aβ metabolic process also had high enrichment factors and moderately low *p* values. Regarding cellular components, six CC clusters were highly enriched in N-exo proteins. Among them, terms including lytic vacuole and brush border membrane were highly enriched with low *p* values. Terms including extracellular membrane and plasma lipoprotein particle were also highly enriched with moderately low *p* values. Regarding molecular functions, the enzyme activity terms including peptidase activity, hydrolase activity, and serine-type exopeptidase activity were highly enriched with low *p* values (Figure 6, Appendix A).

### 2.6. Comparison of the Enriched BP Terms by the Identified Proteins between U-exo and N-exo

Hereafter, the terms of the biological process that were enriched in the U-exo and N-exo proteins were compared. The results showed that 29 enriched terms in U-exo and N-exo overlapped (Figure 7A, Appendix A). Among the 29 overlapped BP terms, six terms including sphingolipid catabolic process, ceramide catabolic process, peptide catabolic process, membrane lipid catabolic process, Aβ clearance, and Aβ metabolic process showed high enrichment (log 2 enrichment factor > 4) in both U-exo and N-exo proteins. Among them, the sphingolipid catabolic process and membrane lipid catabolic process showed the lowest *p* values with high enrichment factors. The term peptide catabolic process was also highly enriched in both U-exo and N-exo, but the *p* value was low only in N-exo. The terms ceramide catabolic process, Aβ clearance, and Aβ metabolic process showed high enrichment with moderately low *p* values (Figure 7B).

### 2.7. Comparison of Exosome Proteins Participating in the Highly Enriched BP Terms in Both U-exo and N-exo

In order to determine the exosomal proteins that play important functions in biological processes, we compared the proteins involved in the five BP terms with high enrichment in both U-exo and N-exo, including sphingolipid catabolic process, membrane lipid catabolic process, ceramide catabolic process, Aβ clearance, and Aβ metabolic process (Figure 7B). The results showed that acid ceramidase was a member of three highly enriched BP terms, sphingolipid catabolic process, ceramide catabolic process, and membrane lipid catabolic process. Importantly, Asah1 significantly increased in both U-exo and N-exo of J20 mice. However, beta-hexosaminidase subunit beta (Hexb) was found to be increased only in N-exo of J20. Acid sphingomyelinase-like phosphodiesterase 3a (Smpdl3a), which is a member of the sphingolipid catabolic process and membrane lipid catabolic process BP terms, was also increased in both U-exo and N-exo. Importantly, alpha-N-acetylgalactosaminidase (Naga) and autotaxin (Enpp2) were detected only in N-exo, whereas sphingomyelin phosphodiesterase (Smpd1) was detected in U-exo, and their levels were increased in J20 compared to WT (Table 4). In Aβ clearance and Aβ metabolic process, A*poE*, clusterin, and neprilysin were detected in both U-exo and N-exo, and their levels were increased in J20. ACE, a member of the Aβ metabolic process, was also increased in J20 (Table 4).

In summary, our results demonstrated that in the early pathology of AD, Asah1 is the key protein in lipid metabolism, and clusterin, A*poE*, neprilysin, and ACE play important roles in Aβ clearance and metabolism.

### 2.8. Protein–protein Interaction (PPI) Analysis

To further explore the association among the proteins identified in U-exo and N-exo, we performed cluster analysis of the PPI network with the molecular complex detection (MCODE) algorithm, and four MCODE complexes were identified in U-exo (Figure 8A) and N-exo (Figure 8B). Notably, MCODE 1 had the highest score (Figure 8C) and contained the members of BP terms Aβ clearance and Aβ metabolic process, including A*poE* and clusterin (Figure 8A,B). In addition, for MCODE1, albumin (Alb), Gas6, immunoglobulin superfamily containing leucine-rich repeat protein (Islr), Cfd, Serpina1d, and Serpina1e were more abundant and were shared in both U-exo and N-exo (Figure 8A,B, Table 1 and Table 2). Pro-epidermal growth factor (Egf) and lactadherin (Mfge8) also existed in MCODE1 of U-exo and N-exo, but they were not different proteins in U-exo and N-exo (Figure 8A,B, Appendix A). For unique proteins, U-exo contains only one non-different protein plasma, alpha-L-fucosidase (Fuca2), in MCODE1 (Figure 5A, Appendix A), whilst N-exo contains three proteins with increased abundance: alpha-1-antitrypsin 1-1 (Serpina1a), ceruloplasmin (Cp), and matrix remodeling-associated protein 8 (Mxra8) in MCODE1 (Figure 8B, Table 2).

### 2.9. Quantification of Clusterin in U-exo and N-exo

Finally, WB analysis was done to verify the efficacy of the iTRAQ-based quantification. Among the important proteins (Asah1, clusterin, A*poE*, neprilysin, and ACE), clusterin has the following important features: (1) Clusterin has markedly higher peptide coverage with 49 and 50 peptides in U-exo and N-exo, respectively (Figure 9A), suggesting that clusterin has the highest confidence and abundance. (2) For GO enrichment analysis of the common different proteins, clusterin was enriched in the BP term regulation of proteolysis (Appendix A). (3) For GO enrichment analysis of the identified proteins in U-exo and N-exo, clusterin was highly enriched in the BP terms Aβ clearance and Aβ metabolic process (Table 4). (4) Clusterin was a member of MCODE complex with the highest score (Figure 8). (5) It was previously reported that clusterin plays an important role in AD and has the potential to become a biomarker of AD [42,43,44]. Considering the limitations of N-exo, we finally chose clusterin for WB analysis. WB results showed that clusterin existed both in U-exo and N-exo and presented as three different molecular weights (about 80, 70, and 50 KDa in size). Interestingly, the band distributions of clusterin in U-exo and N-exo are different. The band at 50 KDa was the major band for N-exo, whereas it was difficult to detect in U-exo (Figure 9B). The levels of the 80 KDa bands were similar between N-exo and U-exo. After normalization, the quantified data showed that all three bands of clusterin were slightly increased in J20 than those in WT in both U-exo and N-exo (Figure 9C).

## 3. Discussion

In this study, we have performed a comprehensive proteomic analysis of U-exo and N-exo based on an AD model mouse at early stage, and a series of proteins were identified in U-exo (79 proteins) and N-exo (117 proteins). Among them, 40 common, different proteins were present in both U-exo and N-exo. Importantly, five biological processes were highly enriched in both U-exo and N-exo. Among them, three biological processes are related to lipid metabolism, including sphingolipid catabolic process, ceramide catabolic process, and membrane lipid catabolic process, and more importantly, the common different protein acid ceramidase was a member of the three BP terms. The other two highly enriched biological processes in both U-exo and N-exo are associated with Aβ, including Aβ clearance and Aβ metabolic process, and their members contain four common different proteins: clusterin, apolipoprotein E, neprilysin, and angiotensin converting enzyme. Thus, our results suggested that in the early stage of AD, acid ceramidase is an important regulator in lipid metabolism, while clusterin, apolipoprotein E, neprilysin, and angiotensin converting enzyme are specific for Aβ pathology. These findings provide new insights into the understanding of the pathology, as well as the investigation of diagnostic biomarkers, of AD that might be valuable for its management.

Clinically, the pathology of AD is very complex. AD can be divided into different types according to age of onset and genetic susceptibility. Sporadic or late-onset AD accounts for more than 95% of cases and begins after the age of 65. Early-onset or familial AD is rare and usually appears before the age of 60. Although familial AD is less common, its associated familial mutations underly major molecular models of the disease. Familial AD-associated mutations are mainly found in components involved in Aβ peptide production, such as APP and presenilin [45,46]. The J20 mouse is a very popular AD mouse model that overexpresses human APP with two mutations linked to familial AD (the Swedish and Indiana mutations), and is widely used for the study of amyloid deposition and AD pathogenesis [47,48,49]. In this study, we are committed to finding specific proteins in the early pathology of AD, so as to reveal the possible pathological mechanism of early AD and lay the foundation for the development of early non-invasive diagnostic markers for AD. In our future studies, clinical urine exosome samples from AD patients will provide more important data to validate the different proteins identified in this study.

Here, we established a method to isolate neuron-derived exosomes from urine according to previous research [50]. Given the high and relatively specific expression of L1CAM in neural tissue and its high expression on exosomes derived from cultured neurons [25], immunoprecipitation with L1CAM antibody is widely used to extract neuron-derived exosomes from blood [51,52,53,54,55,56,57]. The molecular cargo of the exosomes plays a crucial role in intercellular communication in both physiological and pathological conditions and is suggested to reflect the pathophysiological condition of the tissue [58]. Consequently, exosomes have great potential to serve as diagnostic markers in liquid biopsies [59]. Since tissue biopsy in any chronic CNS disease is usually not recommended, exosomal biomarker studies could be an alternative to tissue biopsy here. However, the body fluids used for liquid biopsies will contain exosomes from other sources in addition to CNS. In that case, the enrichment of tissue-specific exosomes is crucial for finding a specific biomarker. Hence, our neuron-specific exosome enrichment system could help to develop a better disease diagnosis system not only for AD but also for other chronic CNS diseases.

The above results proved the success of our exosome isolation.

During the characterization of exosomes, we found that both U-exo and N-exo were positive for exosome markers including Tsg101, ALIX, CD9, and CD63. In addition, TEM shows that both U-exo and N-exo exhibit typical morphology of exosomes, typical spherical bilayer membrane structure, and a diameter range of 30–150 nm [22]. The above results proved our success in isolating exosomes. The quality control analysis of any experimental study is also important. To evaluate the quality, the protein content characteristic analysis in terms of protein subcellular localization was performed. The molecular compositions of exosome cargoes depend on the types and pathophysiological conditions of the cells [60,61]. The cargoes of the exosomes are usually contained in extracellular proteins including cytokines, growth factors, and other signaling proteins. Additionally, membrane proteins are enriched because of their membranous origin. Exosomes also contain cytosol, nuclear, and cytoskeletal proteins. Importantly, organelle-associated proteins are very rare in exosome cargo [58,62,63,64,65]. Our analysis demonstrated that both in U-exo and N-exo, most of the proteins in exosome cargoes are localized in the extracellular regions and membranes, followed by cytosolic proteins. Additionally, all cytoskeletal proteins and organelle-associated proteins only occupy a small portion. Therefore, the protein compositions of both U-exo and N-exo fully conform to the characteristics of exosomes, suggesting a high level of purity in the materials in this study. Moreover, enrichment analysis of cellular components displayed enriched terms closely related with exosomes, such as extracellular exosome, extracellular matrix, brush border membrane, and secretory vesicle, which further supported the high purity of our exosomes. Furthermore, the cargo analysis showed that some of the contents, including α-N-acetylgalactosaminidase, lysosomal α-glucosidase (Gaa), ApoA4, and prion protein (Prnp), are unique to N-exo. Importantly, all of them are known to have roles in CNS diseases [66,67,68,69]. These results indicate that those protein levels are below the detection levels in U-exo. After the enrichment of neuron-derived exosomes by immunoprecipitation, their levels were increased to detection levels, suggesting their CNS source. To assess the degree of CNS origin of N-exo, we checked the brain expression of the identified proteins. The results showed that 30% of U-exo proteins (24 proteins) and 35% of N-exo proteins (41 proteins) could be expressed in the brain. Therefore, compared with U-exo, N-exo contains more proteins that can be expressed in the brain. Strikingly, N-exo also contained fifteen neuronal exosomes, eight of which are unique to N-exo. The above evidence suggested that proteins of neuron origin are better enriched in N-exo, which further supports the effectiveness of immunoprecipitation to isolate neuron-derived exosomes through L1CAM antibody.

To understand the proteomic data, we compared U-exo proteins and N-exo proteins. Several proteins including platelet-activating factor acetylhydrolase, carbonic anhydrase 6, growth arrest-specific protein 6, meprin A subunit beta, neprilysin, and cathepsin B were found to be increased both in U-exo and N-exo. All of these proteins have been implicated in AD pathology [70,71,72,73,74,75]. Some proteins were uniquely increased in one fraction of exosomes, such as cathepsin E and ceruloplasmin levels in N-exo, and cathepsin D in U-exo. The expression of the cathepsin family of proteases was found to be cell-type specific [76]. Since U-exo could derive from a variety of cell types and N-exo from neurons, this could be the reason for such changes in the levels of the proteins, especially cathepsins, in these two types of exosomes. Additionally, cathepsin B activities increased AD pathology, especially in amyloid angiopathy [75,77,78,79], which reflects their changes in J20 mice exosomes. Ceruloplasmin is known to have a role in iron metabolism and redox reactions in the brain [80]. Several studies have shown the link between iron and AD pathology [81,82,83]. Hence, ceruloplasmin could be important for such aberrant iron metabolism in AD conditions. Indeed, a study demonstrated the inhibition of ceruloplasmin increased AD pathology [84]. In this study, we found that ceruloplasmin levels are increased in neuron-derived exosomes, suggesting that increased ceruloplasmin production in this mouse model could be a protective mechanism that is involved in the control of the redox state.

For further analysis of the data concerning functional biological processes, gene ontology enrichment analysis was performed. In the analysis, the genes associated with the proteins are hierarchically clustered under functional terms, which describe the biological process, molecular function, or cellular component, and translate the raw information into meaningful pathophysiological knowledge. In the biological process category, the proteins in U-exo and N-exo are mostly clustered under the terms of lipid metabolic processes, amyloid β metabolism, and clearance processes. In the lipid metabolic process, the term ‘sphingolipid catabolic process’ was highly enriched. The levels of several proteins clustered under that term were increased in J20 exosomes. Among them, acid ceramidase levels were increased in both U-exo and N-exo. The function of this enzyme is to activate the conversion of ceramide to sphingosine and to concomitantly increase sphingosine-1-phosphate (S-1-P) secretion [85,86,87,88]. The S-1-P in turn acts as a signaling molecule and activates AKT signaling [89,90]. The importance of such signaling in AD has been described in several reports [91,92,93]. Since the levels of acid ceramidase are increased in both in U-exo and N-exo, such altered lipid metabolism could occur in other organs along with the brain in this AD model. Additionally, autotaxin was shown to be uniquely increased in J20 N-exo. Autotaxin is known to increase S-1-P as well as lysophosphatidic acid [94]. Both of these are signaling lipids and are implicated in AD pathology [95,96]. Hence, sphingolipid metabolisms could be important for AD pathology.

Additionally, in the case of the biological process category, the common different proteins in both U-exo and N-exo were enriched in 13 terms, and most of the terms are related to protein metabolism, especially the regulation of enzyme activity. Notably, only 18 common different proteins (Pzp, Ambp, Ctsb, Gas6, Plau Serpina1d, Serpina1e, Serpina3k, A*poE*, clusterin, neprilysin, ACE, Anpep, Klk1, Mep1a, Actg1, Cfd, and Mup1) were involved in the 13 terms. Interestingly, both the total identified U-exo and N-exo proteins were highly enriched in Aβ clearance and Aβ metabolic process by U-exo whose numbers also contain the four common different proteins A*poE*, clusterin, neprilysin, and ACE, suggesting that the two processes related to Aβ are generally altered in this mouse model. A*poE* helps to clear Aβ through perivascular pathways [97], and clears clusterin or ApoJ through low density lipoprotein receptor-related protein 2 [98]. Since both of them are increased in J20 exosomes and can bind Aβ [98], their increased levels may help to accumulate the peptide after binding and enhance exosome-mediated clearance. The metallopeptidase enzyme neprilysin levels were also increased along with ACE, which is also enriched under the term Aβ metabolism. Hence, it is possible that Aβ is trapped by A*poE* and clusterin in exosomes, and neprilysin [74] and ACE [99] help to degrade it and clear over-produced Aβ from this APP transgenic mouse brain. Such increased levels of Aβ degradation enzymes along with lipid metabolism enzymes caused enrichment enzyme activity terms under the molecular function category.

Finally, we have performed a protein–protein interaction analysis to identify the partner proteins that are important in AD pathological processes. Usually, proteins can hardly function as isolated species in the body, and over 80% of proteins do not function individually but in complexes [100]. Among four identified MCODE complexes, MCODE 1 with the highest MCODE score is the most significant module in the PPI network [101,102]. Remarkably, the common different proteins clusterin and A*poE* as members of MCODE 1 were highly enriched in the BP terms Aβ clearance and Aβ metabolic process in both U-exo and N-exo, further suggesting the significant importance of MCODE 1 in the early pathology of AD. The analysis identified several proteins, including serpinas, clusterin, A*poE*, growth arrest-specific protein 6, and albumin, in the MCODE 1 and they all increased in both U-exo and N-exo. All of them, including serpinas, have been shown to have important functions in Aβ metabolism or clearance [97,98,103,104,105]. Interestingly, matrix remodeling-associated protein 8 (Mxra8) showed partnering with these proteins and elevated levels in N-exo. The downregulation of Mxra8 could induce ferroptosis by elevating Fe^2+^ [106]. Iron accumulation promotes Aβ deposition and toxicity [107,108]. It will be interesting to investigate the role of Mxra8 in the clearance or degradation of Aβ in AD conditions. In MCODE 1, complement factor D as an interactor of A*poE* is also increased in J20 U-exo and N-exo. Notably, in addition to Aβ clearance and Aβ metabolic process, complement factor D, A*poE*, and ACE were enriched in vasculature development. Previous studies have concluded that cerebral vascular dysregulation is the earliest and strongest pathological factor associated with late-onset AD [109,110]. A*poE* was proved to promote Aβ clearance in human blood vessels [97]. Thus, we hypothesize that A*poE* interacts with complement factor D to clear Aβ through perivascular pathways, which is worth exploring further in the future.

Though our study revealed several key pathological mechanisms of AD pathology, some limitations need to be mentioned. Firstly, we used only an early time point (3 months) to check the early changes in the pathology. However, such a one-time-point-study could show a partial picture of the pathology. A time-dependent study would provide more detailed information, which would be helpful to select a disease diagnosis marker in urine samples. Secondly, when analyzing urine exosome protein using bioinformatics tools, we identified several pathways related to lipids or Aβ metabolisms that are perturbed in J20 mice. Investigation of the actual regulation of these pathways in the brains of J20 mice was beyond the scope of this study. A detailed molecular investigation related to these pathway changes in the brain could further strengthen our bioinformatic findings and provide valuable information regarding AD pathology. Then, a study with the urine of human AD subjects could be undertaken to identify the candidate marker that has usefulness for community screening of the disease. Thirdly, we studied the proteomics of exosome proteins isolated from urine. However, we did not check the status of the kidney in these mice. A study to check the protein levels in exosomes and correlate them with kidney status would be interesting.

In conclusion, we have established a method of purification of neuron-derived exosomes from urine, performed a comprehensive proteomics analysis, and demonstrated the enrichment of AD-related proteins in both urine and neuron-derived exosomes. Such an analysis is important to understand the pathophysiology of the disease, to identify new therapeutic targets as well as to identify novel diagnostic markers for the disease.

## 4. Materials and Methods

Workflow of our experiment was shown in Figure 10.

### 4.1. Animals and Samples

Male mice with the human amyloid precursor protein (hAPP) transgene (J20) were used as an AD model and their wild-type (WT) littermates were used as a control. J20 mice overexpress the human APP transgene that carries two mutations, Swedish (K670N/M671L) and Indiana (V717F), which are associated with familial AD. J20 mice express more Aβ than mice with wild-type human APP [47,48,49]. J20 mice initially exhibited a mild AD phenotype at 1-month-old, showing only Aβ puncta deposition in the hippocampus [111]. Additionally, the AD phenotype gradually became obvious over time. By 3 months, the number of neurons in J20 mice decreased in the CA1 region [47], and J20 mice showed basic synaptic transmission defects [48] and synapse loss [111]. By 4 months, J20 mice exhibited deficits in spatial reference memory [47,49]. Remarkably, J20 mice developed robust amyloid plaques at 5 to 7 months of age [112].

Urine was isolated from J20 mice and WT mice at 3 months of age. To obtain uncontaminated, reliable samples, we used metabolic cages to collect mouse urine, one mouse per cage. We used a total of 18 mice, including 9 J20 mice and 9 WT mice. Mouse urine was collected every 24 h for 1 month from 12 to 16 weeks. For one mouse, about 0.5–1 mL of urine was collected per 24 h. J20 mice produced less urine per day than WT mice. Thus, for one wild-type mouse, we collected a total of about 20 mL of urine in one month. For one J20 mouse, we collected a total of about 15 mL of urine in one month. Mice were cultured in a specific pathogen-free environment at a temperature of 25 °C with 12 h of light per day, and the mice were free to eat and drink. The collected mouse urine was stored at −80 °C until the experiment. All of the experiment programs involved in this study were approved by the Ethical Committee of Shimane University School of Medicine (approval number: IZ29–28) and followed the experimental principles.

### 4.2. Exosome Isolation

In order to obtain sufficient exosomal proteins, we used pooled samples. After urine collection, exosomes were isolated using the miRCURY^®^ Exosome Kits (Qiagen, Hilden, Germany). Then, pooled samples of 3 mice were used for iTRAQ of U-exo. The N-exo was extracted from U-exo by immunoprecipitation; therefore, the volume is less, and we used the pooled samples of 9 mice for N-exo isolation and iTRAQ. In addition, the pooled samples of 9 mice were also used for TEM and WB.

We used miRCURY^®^ Exosome Kits to isolate exosomes according to the manufacturer’s instructions. Briefly, collected mouse urine specimens were thawed on ice. Residual dead cells were removed by centrifugation at 3000× *g* and 4 °C for 10 min. Cell debris, that included large EVs (apoptotic body and microvesicle), was further removed by a simple filtration using a 0.22 μm filter (Millipore, Billerica, MA, USA). An amount of 10 mL of supernatant was collected to mix with 4 mL of precipitation buffer and incubated at 4 °C for 1 h. After this, the sample was centrifuged at 3200× *g* for 30 min at 20 °C. In the end, the supernatant was completely removed and the exosome pellet was dissolved in PBS (pH 7.4) for immunoprecipitation or dissolved in RIPA buffer for WB.

For immunoprecipitations, anti-L1CAM antibody (Sino Biological, Beijing, China) was used to precipitate N-exo from U-exo according to a previous publication with some modifications [50]. In brief, 500 μg of U-exo in 1 mL of PBS (pH 7.4) was incubated at 4 °C for 6 h with 2 μg of rabbit anti-mouse L1CAM antibody. Then, 20 μL of Protein G-agarose beads (Santa Cruz Biotechnology, Dallas, TX, USA) was added into the sample and incubated with gentle rotation overnight at 4 °C. The next day, after centrifugation at 3000× *g* for 5 min at 4 °C and removal of the supernatant, a pellet (L1CAM^+^ N-exo) was suspended in 200 μL of 0.1 M glycine-HCl (pH 2.5–3.0) and then strongly vortexed for 30 s. After centrifugation at 4500× *g* for 5 min at 4 °C, the supernatant containing exosomes were collected and neutralized by 15 μL Tris-HCl (pH 8.0).

### 4.3. Exosomal Protein Isolation

To extract the total protein of exosomes for WB, the exosome pellets were lysed in RIPA buffer (1% Nonidet P 40, 0.5% sodium deoxycholate, and 0.1% sodium dodecyl sulfate in PBS) supplemented with phenyl methyl sulphonyl fluoride and aprotinin for 1 h on ice, and then the lysed sample was sonicated by 25–30 pulses at medium power for 2–3 min in order to dissolve completely. At last, exosome lysates were centrifuged at 12,000 rpm for 10 min at 4 °C to remove exosomal debris.

For iTRAQ, 50 μL of exosome in PBS was added to 200 μL of lysis buffer (500 mM of triethylammonium bicarbonate (TEAB) containing 7 M urea and 0.1% Nonidet P 40) in order to isolate exosome protein, and then the lysed sample was sonicated with Bioruptor (Bioruptor UCD-200TM, COSMO BIO, Tokyo, Japan) for 3 min at medium power. Afterwards, the sample was mixed by rotator at maximum speed at 4 °C for 2 h. Ultimately, a centrifugal filter of 3 KDa (Millipore, Billerica, MA, USA) was used to change the sample buffer from lysis buffer to 50 mM TEAB.

Protein concentration was determined by a BCA assay (Takara, Kusatsu, Japan).

### 4.4. Western Blotting (WB)

For exosome marker and clusterin detection by WB, we isolated the exosome from the equivalently pooled urine of 9 mice for every group. Equivalent protein (40 μg per lane) of exosome lysates was separated by 10% sodium dodecyl sulfate polyacrylamide gel electrophoresis. After being electro-transferred to an activated polyvinylidene fluoride membrane (Millipore, Billerica, MA, USA) at 100 V for 60–100 min, the samples were blocked with 5% skimmed milk for 1 h at room temperature. After being washed 3 times using tris buffered saline containing 0.05% Tween-20 (TBST), the membranes were incubated with polyclonal rabbit anti-mouse Tsg101 (1:500 dilution in TBST; GeneTex, Irvine, CA, USA), monoclonal mouse anti-mouse ALIX (1:200 dilution in TBST; Santa Cruz Biotechnology, Dallas, TX, USA), monoclonal mouse anti-mouse CD9 (1:200 dilution in TBST; Santa Cruz Biotechnology, Dallas, TX, USA), monoclonal mouse anti-mouse CD63 (1:200 dilution in TBST; Santa Cruz Biotechnology, Dallas, TX, USA), and polyclonal rabbit anti-mouse clusterin (1:300 dilution in TBST; Bioss Inc., Boston, MA, USA) overnight at 4 °C. The blots were then washed three times using TBST and incubated with goat anti-rabbit IgG antibody conjugated with peroxidase (1:5000 dilution in TBST; Sigma-Aldrich, St. Louis, MO, USA) and mouse IgG Fc binding protein conjugated to horseradish peroxidase (1:1000 dilution in TBST; Santa Cruz Biotechnology; USA) as secondary antibodies for 1 h at room temperature. Subsequently, the blots were washed three times using TBST and the signals were detected using the AmershamTM ECL Western Blotting Detection Reagent (Amersham Sciences, Amersham, Britain).

### 4.5. Transmission Electron Microscopy (TEM)

Exosome samples from the equivalently pooled urine of 9 mice for every group were subjected to uranyl acetate negative staining. An amount of approximate 10 µL of freshly isolated exosome sample was dropped onto a 400-mesh carbon-coated grid. After incubation at room temperature for 1 min, an additional approx. 10 μL of 2% uranyl acetate was dropped onto the grid and incubated for 1 min. Following this, the grid edges were dried with filter paper. The morphological features of the exosomes were visualized with TEM (Topcon EM-002B, Tokyo, Japan).

### 4.6. Denaturation, Reduction, Cysteine Alkylation, Trypsin Digestion, iTRAQ Labeling, and Strong Cation Exchange Chromatography

The isolated exosomes from the equivalently pooled urine (3 mice for U-exo and 9 mice for N-exo) was used for iTRAQ with 3 replications for U-exo and no replications for N-exo.

Sample preparation was performed according to the introductions provided by AB Sciex (Foster, CA, USA) and previous papers [113,114]. For each group at each stage, 50 μg of filtered exosome protein in 20 μL of 50 mM TEAB was incubated with 1 μL of denaturation reagent (2% SDS dissolved in 50 mM TEAB) and 2 μL of reduction reagent (50 mM tris-(2-carboxyethyl) phosphine (TCEP) dissolved in 50 mM TEAB) at 60 °C for 1 h. Then, the sample was alkylated with 1 μL of cysteine blocking reagent (200 mM methyl methanethiosulfonate (MMTS) dissolved in isopropanol) at room temperature for 10 min. After reduction and alkylation, the sample was digested with trypsin (1:30 *w*/*w*, AB Sciex) overnight at 37 °C. Each digestion was labeled with a different iTRAQ tag by an iTRAQ Reagent Multiplex kit (AB Sciex). The iTRAQ label was diluted 5 times by absolute ethanol, and then a diluted 15 μL of iTRAQ label (114 for WT and 115 for J20) was added to the sample, followed by 60 μL of absolute ethanol. After incubation for 1 h at room temperature, we mixed the two samples. After that, the combined samples were fractionated into six fractions by SCX chromatography according to the manufacturer’s instructions (AB Sciex). Afterwards, each fraction was desalted by a Sep-Pak C 18 cartridge according to the manufacturer’s instructions (Waters, Milford, MA, USA).

### 4.7. Nano-LC Spotting

One fraction from SCX chromatography was further fractionated to 171 spots with a DiNa nanoLC system (KYA Technologies, Tokyo, Japan) while mixing directly with a matrix (4 mg/mL alpha-cyano-4-hydroxycinnamic acid (CHCA); Wako, Osaka, Japan) according to the manufacturer’s instructions and previous publications [113,115]. Later, the spots were collected on an Opti-TOF LC/MALDI 384 target plate (AB Sciex) using a DiNa MaP fraction collector (KYA Technologies) according to the manufacturer’s instructions and previous papers [116].

### 4.8. MALDI-TOF MS/MS Analysis

A mass spectrometer 5800 MALDI-TOF/TOF analyzer (AB Sciex) with TOF/TOF Series software (version 4.0) was used for MS acquisition of spectra between m/z 800 and 4000 in positive ion mode. Parent ions of des-Arg1-bradykinin, angiotensin I, Glu1-fibrinopeptide B, adrenocorticotrophic hormone (ACTH) (clip 1–17), ACTH (clip 18–39), and ACTH (clip 7–38), diluted in a matrix (4 mg/mL CHCA), were used for calibration. Monoisotopic precursor selection for MS/MS was performed by automatic precursor selection with an interpretive method using the DynamicExit algorithm (AB Sciex). MS/MS data acquired from the 5800 MALDI-TOF/TOF were analyzed by ProteinPilot™ software (version 3.0) with the Paragon Protein Database Search Algorithm (AB Sciex). The statistical analyses for iTRAQ analysis were based on ProteinPilot™ software (version 3.0). Each MS/MS spectrum was searched against the database built by AB Sciex (version 20081216, 20,489 entries). Quantitative changes for each protein between the two groups were calculated based on the iTRAQ ratio of the J20 group to the WT group for each peptide. Search results were filtered by a decoy search strategy utilizing a reverse database and a global false discovery rate (FDR) of 5% [50,115]. The statistical method of iTRAQ analyses followed ProteinPilot 3.0 software (AB Sciex) using the Paragon protein database search algorithm. Each MS/MS spectrum was searched against the database (BR_ITRAQ4Plex_Peptide_Labeled_17 and BR_ITRAQ4_Peptide_(mouse)_RQ2017) constructed by AB Sciex (version 20081216; 20,489 entries).

### 4.9. Bioinformatics Analysis

We submitted the total proteins identified in U-exo and N-exo to Metascape (https://metascape.org/ accessed on 4 June 2022) to conduct GO pathway enrichment analysis. Here, pathway enrichment was measured by *p* value and enrichment factor. The *p* values were calculated based on the accumulative hypergeometric distribution. The enrichment factor was calculated by dividing the hit ratio by the background ratio. The hit ratio represents the ratio of the number of genes in the enriched pathway to the total number of genes used for analysis. The background ratio represents the ratio of the number of annotated genes in the enriched pathway to the number of all background genes. The enrichment factor indicates how many times more a given pathway member is found in our gene list compared to what would have been expected by chance. The value range of the *p* value was [0, 1]. The closer it was to zero, the more significant the enrichment was. Pathways with *p* < 0.05 were defined as pathways that were significantly enriched. The larger the enrichment factor, the greater the degree of enrichment.

To further capture the relationships between the enriched terms, hierarchical clustering analysis on the GO enriched terms was performed. A subset of enriched terms was selected and rendered as a network plot. The enriched terms were collected and grouped into clusters based on their membership similarities. The kappa scores were used as the similarity metric, and sub-trees with a similarity of >0.3 were considered a cluster [117]. The most statistically significant term (smallest *p* value) within a cluster was selected to represent the cluster to avoid wasting time on redundant enriched terms.

All pathway enrichment analyses adopted uniform criteria, a *p* value < 0.001, a minimum count of 3, and an enrichment factor > 1.5 to filter out pathways that were not enriched enough.

Metascape was also used to carry out PPI enrichment analysis with the following databases: STRING, BioGrid, OmniPath, and InWeb_IM. Only physical interactions in STRING (physical score > 0.132) and BioGrid were used. The resultant network contained the subset of proteins that form physical interactions with at least one other member in the list. If the network contained between 3 and 500 proteins, the MCODE algorithm was applied to identify densely connected network components [118]. The MCODE score of this network component reflected the stability and importance in the network.

In addition, the subcellular localization of identified proteins was carried out by the online tool WoLF PSORT (https://psort.hgc.jp/, accessed on 27 August 2022). The brain expression of the identified proteins was performed by GenAtlas (http://genatlas.medecine.univ-paris5.fr/ accessed on 20 August 2022). The online database CellMarker 2.0 (http://117.50.127.228/CellMarker/ accessed on 10 September 2022) was used to search for neuronal markers. Bar graphs of GO enrichment analysis were generated by an online tool (http://www.bioinformatics.com.cn/ accessed on 11 July 2022) with R package, and dot bubble plots of enriched BP terms were also created by the R package.

## Figures and Tables

**Figure 1 ijms-24-00672-f001:**
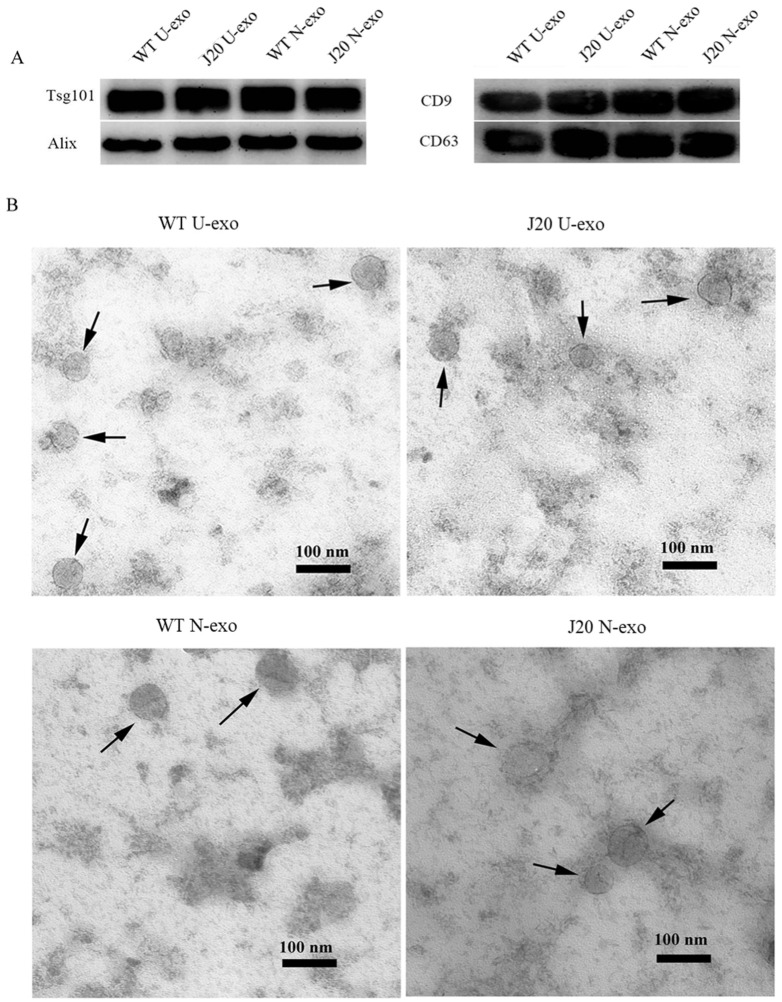
Characterization of isolated exosomes. (**A**) Western blotting for detecting the exosomal markers of U-exo and N-exo. (**B**) Transmission electron microscope for imaging of U-exo and N-exo. U-exo: urine exosomes. N-exo: neuron-derived exosomes.

**Figure 2 ijms-24-00672-f002:**
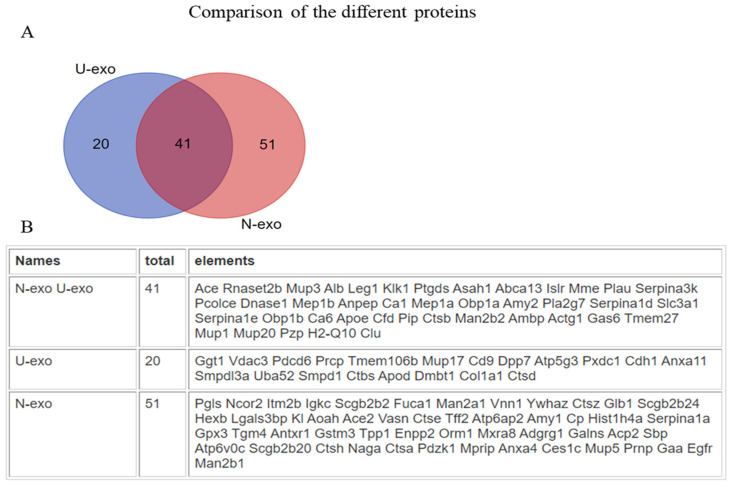
The comparison of the different proteins we identified in U-exo and N-exo. (**A**) Venn diagram reflecting the identified different proteins in U-exo and N-exo and (**B**) corresponding proteins. U-exo: urine exosomes. N-exo: neuron-derived exosomes.

**Figure 3 ijms-24-00672-f003:**
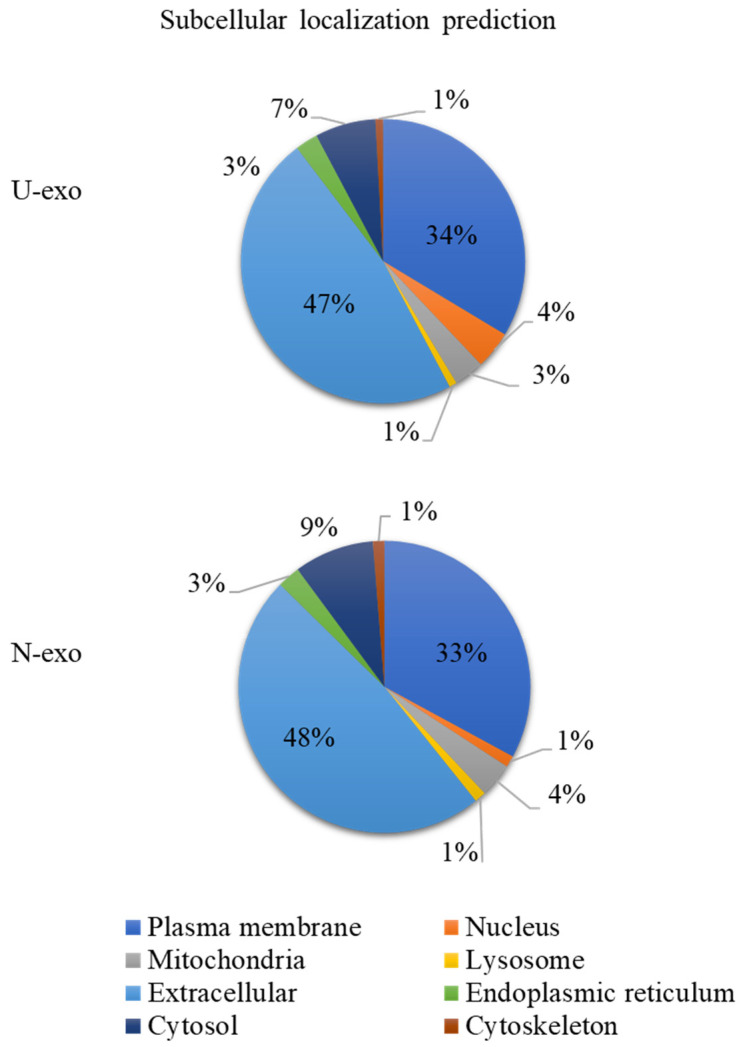
Pie chart of subcellular localization of identified proteins. The online tool WoLF PSORT (https://psort.hgc.jp/ accessed on 27 August 2022) was used to conduct subcellular localization of identified proteins. According to species and multifasta format protein sequence, WoLF PSORT will calculate scores for possible localization sites. The larger the value, the more likely the protein is localized at the position. Only the subcellular location with the highest score was used to generate the percentage pie chart of subcellular localization (Appendix A for details). U-exo: urine exosomes. N-exo: neuron-derived exosomes.

**Figure 4 ijms-24-00672-f004:**
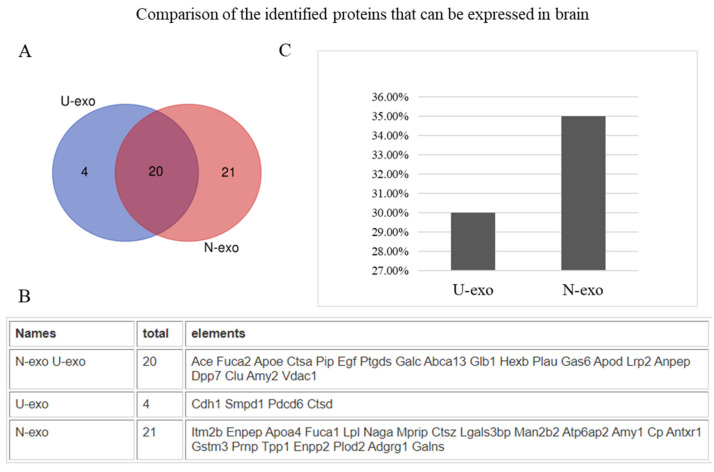
The comparison of the identified proteins which can be expressed in the brain in both U-exo and N-exo. (**A**) Venn diagram showing the identified proteins which can be expressed in the brain in both U-exo and N-exo and (**B**) corresponding proteins. (**C**) The percentage of the identified proteins which can be expressed in the brain to the total identified proteins. U-exo: urine exosomes. N-exo: neuron-derived exosomes.

**Figure 5 ijms-24-00672-f005:**
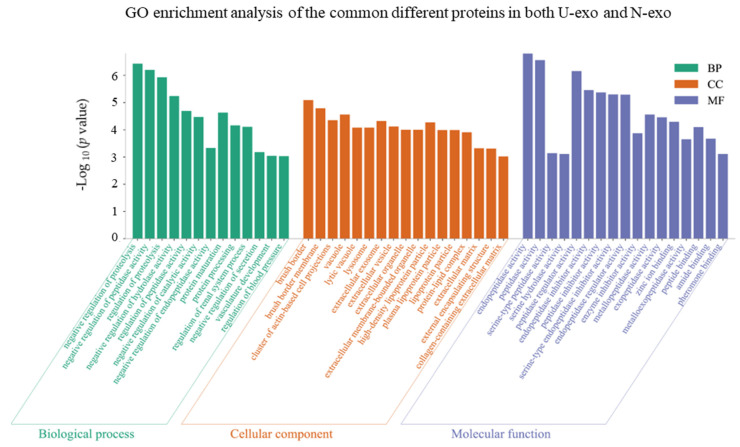
GO enrichment analysis of the common different proteins identified in both U-exo and N-exo. U-exo: urine exosomes. N-exo: neuron-derived exosomes.

**Figure 6 ijms-24-00672-f006:**
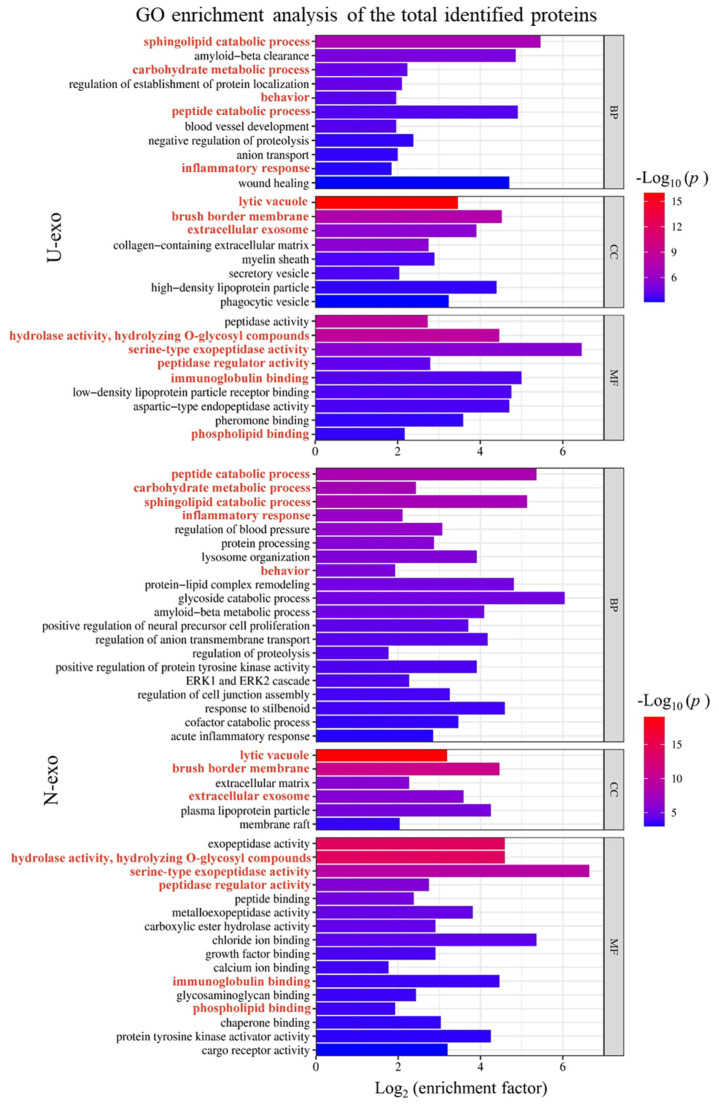
Bar graph for the gene ontology enrichment analysis of the total identified proteins in U-exo and N-exo. We carried out hierarchical clustering analysis on the enriched terms of the identified proteins in U-exo and N-exo. The most statistically significant term (smallest *p* value) within a cluster is chosen to represent the cluster. Pathway enrichment is measured by *p* value and enrichment factor. The smaller the *p* value, the larger the value of the enrichment factor, and the higher the enrichment degree of the corresponding term. The bar color is displayed in a gradient from red to blue, and the enriched BP terms were ranked in descending order of the −Log10 (*p* value). The length of the bar represents Log2 (enrichment factor). BP: biological process. CC: cellular component. MF: molecular function. U-exo: urine exosomes. N-exo: neuron-derived exosomes.

**Figure 7 ijms-24-00672-f007:**
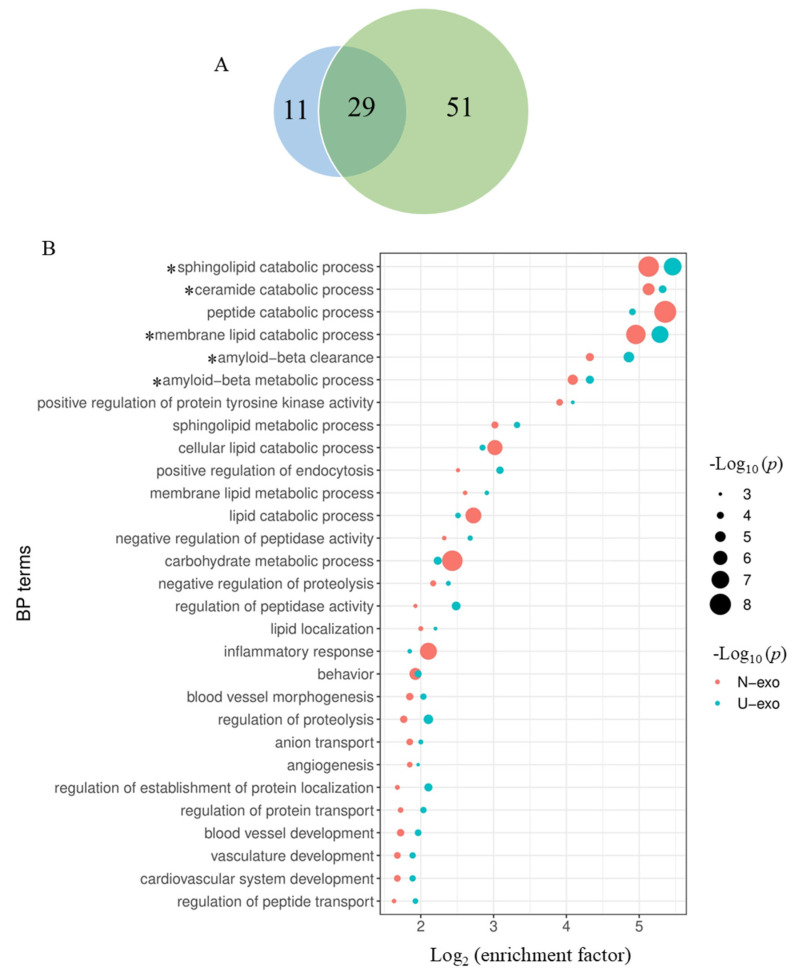
Comparison of the BP enrichment analysis between the identified proteins of U-exo and N-exo. (**A**) Venn diagram showing the enriched BP terms of the identified proteins in U-exo and N-exo. The blue area represents the enriched BP terms of the identified proteins in U-exo, while the green area represents the enriched BP terms of the identified proteins in N-exo. (**B**) Dot bubble plot for displaying the overlapped enriched BP terms between U-exo and N-exo. Pathway enrichment is measured by *p* value and enrichment factor. Node color represents the sample group (red represents the N-exo group and blue represents the U-exo group). The size of the nodes is arranged in descending order of the −log10 (*p* value). The x-axis represents log2 (enrichment factor). The enriched BP terms were ranked in descending order of the value of log2 (enrichment factor). * represent important enriched BP pathways for further analysis. BP: biological process. U-exo: urine exosomes. N-exo: neuron-derived exosomes.

**Figure 8 ijms-24-00672-f008:**
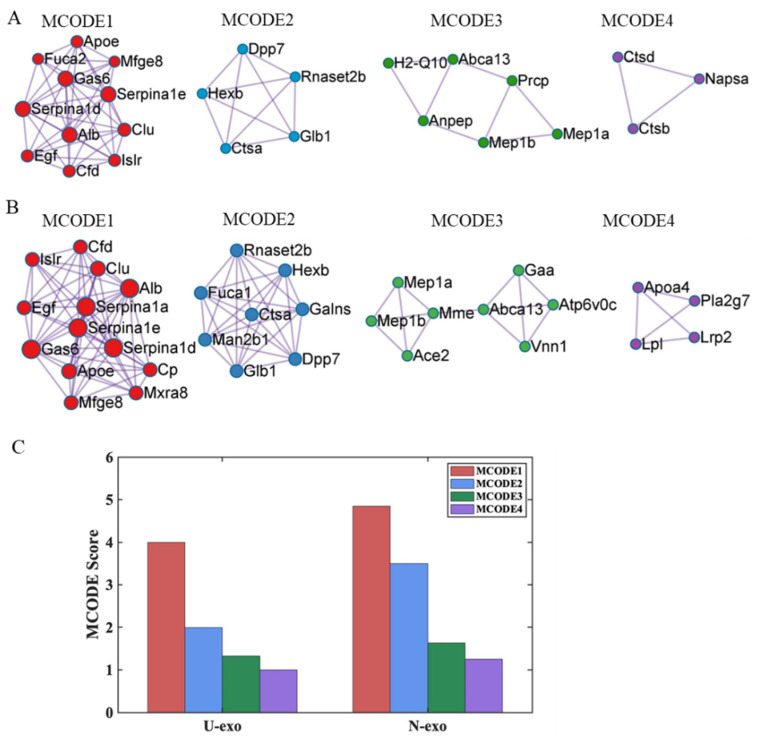
Protein–protein interaction analysis of the identified proteins in U-exo and N-exo. MCODE components identified from the identified protein lists in (**A**) U-exo and (**B**) N-exo. The MCODE algorithm was applied to identify densely connected network components. (**C**) The MCODE score value of identified MCODE components. The visualization of all MCODE components was generated with Cytoscape 3.8.0. Colors show the different components of MCODE (red color: MCODE 1, blue color: MCODE 2, green color: MCODE 3, and purple color: MCODE 4). MCODE: molecular complex detection.

**Figure 9 ijms-24-00672-f009:**
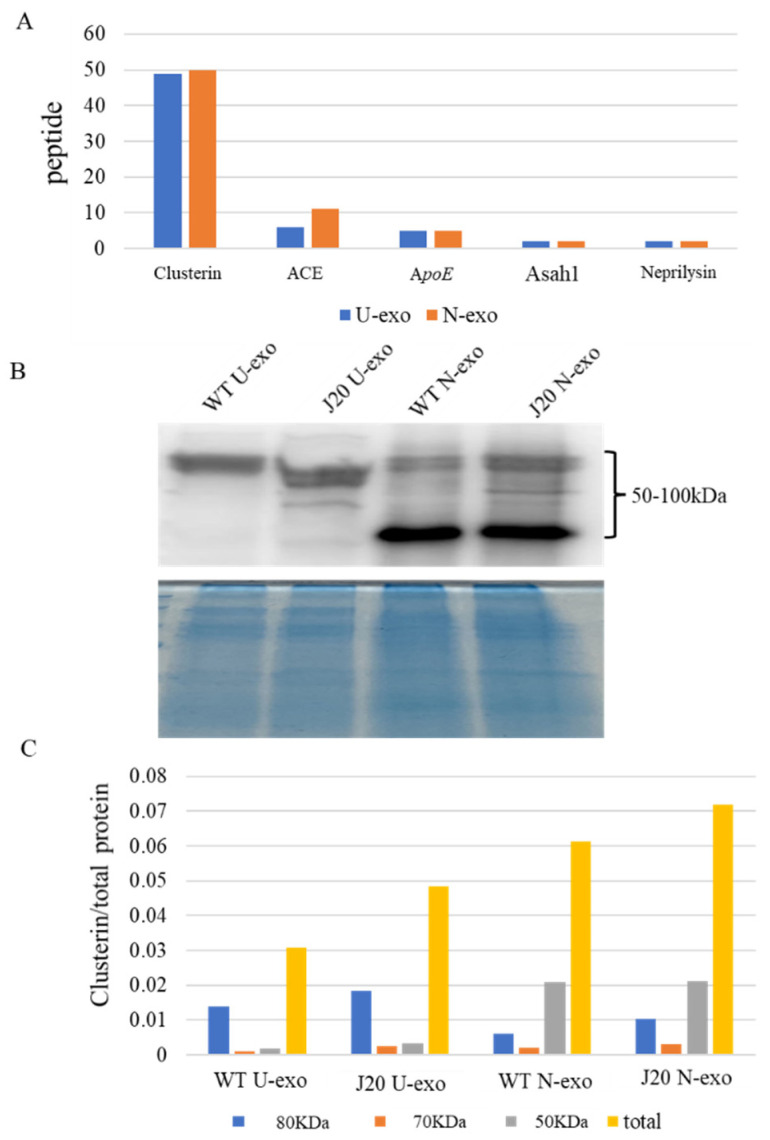
Validation of the most important protein. (**A**) Comparison of the number of distinct peptides with at least 95% confidence among screened important proteins in the early stages of AD pathology. (**B**) Western blotting for clusterin. The total protein stained by Coomassie brilliant blue (CBB) was used as loading control. (**C**) Relative levels of clusterin among groups. Clusterin was normalized by the total protein stained by CBB. U-exo: urine exosomes. N-exo: neuron-derived exosomes.

**Figure 10 ijms-24-00672-f010:**
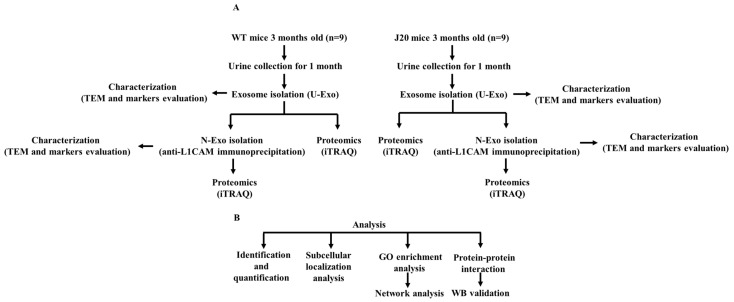
Workflow of experiment design. (**A**) Isolation, characterization, and proteomics of urine exosomes (U-exo) and neuron-derived exosomes (N-exo) in male mice with the human amyloid precursor protein (hAPP) transgene (J20) and their wild-type (WT) littermate control. First, the U-exo was isolated from urine by polymer precipitation, then N-exo was isolated from U-exo by immunoprecipitation with the anti-neural cell adhesion molecule L1 (L1CAM) antibody. Second, the two exosomes were characterized by transmission electron microscope (TEM) and Western blotting (WB). Lastly, the protein profiles of U-exo and N-exo were identified using isobaric tags for relative and absolute quantitation (iTRAQ)-based matrix assisted laser desorption ionization time of flight mass spectrometry/mass spectrometry (MALDI TOF MS/MS). (**B**) Bioinformatics analysis and protein validation. The identified proteins were analyzed by bioinformatic analysis. Then, the most important proteins screened by bioinformatics were validated by WB.

**Table 1 ijms-24-00672-t001:** The different proteins we identified in U-exo between WT and J20 groups.

Unused ProtScore ^a^	% Coverage ^b^	Peptides (95%) ^c^	UniProt Accession	Gene Symbol	Name	Mean Ratio ^d^
3.15	9.5	1	Q80 × 71	Tmem106b	Transmembrane protein 106B	2.6911
14	47.2	8	Q9D3H2	Obp1a	Odorant-binding protein 1a	2.5735
6.38	17.4	3	P18761	Ca6	Carbonic anhydrase 6	2.4163
2.35	15.6	1	Q8JZU6	Pxdc1	PX domain-containing protein 1	2.4042
8	17.2	4	P13634	Ca1	Carbonic anhydrase 1	2.3034
6.54	32.2	4	A2AEP0	Obp1b	Odorant-binding protein 1b	2.2020
4.26	12.3	2	P01898	H2-Q10	H-2 class I histocompatibility antigen, Q10 alpha chain	1.9817
89.56	79.1	75	P00688	Amy2	Pancreatic alpha-amylase	1.9728
44.52	48.7	23	P07724	Alb	Serum albumin	1.9404
5.49	13.4	2	Q60963	Pla2g7	Platelet-activating factor acetylhydrolase	1.8847
19.33	39	10	Q07456	Ambp	Protein AMBP	1.8396
14.02	11.4	8	Q60997	Dmbt1	Deleted in malignant brain tumors 1 protein	1.8363
10.55	54.5	24	Q00898	Serpina1e	Alpha-1-antitrypsin 1-5	1.8309
15.5	36.5	11	Q8C6C9	Leg1	Protein LEG1 homolog	1.8007
3.52	87.2	371	P11588	Mup1	Major urinary protein 1	1.7617
43.75	63	32	Q00897	Serpina1d	Alpha-1-antitrypsin 1-4	1.7579
14.5	34.5	10	P07759	Serpina3k	Serine protease inhibitor A3K	1.7509
13.25	89.5	116	Q5FW60	Mup20	Major urinary protein 20	1.6810
4	6.8	2	Q61391	Mme	Neprilysin	1.6804
10.8	52.7	5	P02816	Pip	Prolactin-inducible protein homolog	1.6792
4.7	22.4	2	P10605	Ctsb	Cathepsin B	1.6491
2.18	6.8	1	Q6GU68	Islr	Immunoglobulin superfamily containing leucine-rich repeat protein	1.6337
5.4	30.2	4	P51910	Apod	Apolipoprotein D	1.5999
14.96	17.2	6	P09470	Ace	Angiotensin-converting enzyme	1.5564
9.02	9.2	4	P56384	Atp5g3	ATP synthase F(0) complex subunit C3	1.5092
9.02	9.2	4	Q5SSE9	Abca13	ATP-binding cassette sub-family A member 13	1.5018
21.68	40.6	21	P15947	Klk1	Kallikrein-1	1.4949
4.38	11.2	2	C0HKG6	Rnaset2b	Ribonuclease T2-B	1.4947
2	10.2	1	P06869	Plau	Urokinase-type plasminogen activator	1.4821
4	27.9	3	Q60931	Vdac3	Voltage-dependent anion-selective channel protein 3	1.4567
4	4.4	2	O54782	Man2b2	Epididymis-specific alpha-mannosidase	1.4522
6.22	14.9	3	P97384	Anxa11	Annexin A11	1.4433
10.11	19.3	5	P08226	Apoe	Apolipoprotein E	1.4288
12.42	9.8	5	Q61838	Pzp	Pregnancy zone protein	1.4280
18.4	46.1	12	P49183	Dnase1	Deoxyribonuclease-1	1.4277
3.53	12.4	3	P18242	Ctsd	Cathepsin D	1.4262
12.14	32.8	7	O09114|	Ptgds	Prostaglandin-H2 D-isomerase	1.4217
4.26	7.1	2	Q61592	Gas6	Growth arrest-specific protein 6	1.3947
4.07	9.6	2	Q8R242	Ctbs	Di-N-acetylchitobiase	1.3938
9	12.4	4	P70158	Smpdl3a	Acid sphingomyelinase-like phosphodiesterase 3a	1.3852
133.09	88	230	P04939	Mup3	Major urinary protein 3	1.3838
12.55	19.2	8	Q61847	Mep1b	Meprin A subunit beta	1.3830
3.71	15.5	2	Q9WV54	Asah1	Acid ceramidase	1.3824
0.8	5.3	0	P40240	Cd9	CD9 antigen	1.3763
23	87.2	367	B5X0G2	Mup17	Major urinary protein 17	1.3637
8.11	10.2	4	P09803	Cdh1	Cadherin-1	1.3598
24.08	47.1	17	P03953	Cfd	Complement factor D	1.3478
4.04	12.6	2	P12815	Pdcd6	Programmed cell death protein 6	1.3386
2.01	9.6	1	Q04519	Smpd1	Sphingomyelin phosphodiesterase	1.3335
13.79	27.5	7	P63260	Actg1	Actin, cytoplasmic 2	1.3288
60.93	60.7	49	Q06890	Clu	Clusterin	1.3055
2.2	11.7	1	Q9ESG4	Tmem27	Collectrin	1.3012
23.72	33	15	Q9ET22	Dpp7	Dipeptidyl peptidase 2	1.2554
16.26	22.7	8	Q60928	Ggt1	Gamma-glutamyltranspeptidase 1	1.2462
14.17	21.3	7	Q91WV7	Slc3a1	Neutral and basic amino acid transport protein rBAT	1.2417
12.29	13.9	7	P97449	Anpep	Aminopeptidase N	1.2401
4	15.2	2	Q61398	Pcolce	Procollagen C-endopeptidase enhancer 1	1.2320
8	14.5	4	Q7TMR0	Prcp	Lysosomal Pro-X carboxypeptidase	1.2307
148.27	56.1	175	P28825	Mep1a	Meprin A subunit alpha	1.2239
10.59	20.4	5	P11087	Col1a1	Collagen alpha-1(I) chain	1.2221
6.01	44.5	3	P62984	Uba52	Ubiquitin-60S ribosomal protein L40	1.2113

Table 1 shows the different proteins between J20 and WT meeting the following criteria: (1) FDR < 5%. (2) The number of distinct peptides with at least 95% confidence ≥ 1. (3) The protein can be detected with ratio at least two times in three biological replicates. (4) Fold change of 1.2. ^a^: Unused ProtScore is a score of protein confidence, and reflects the amount of total and unique peptide evidence related to an identified protein. Unused ProtScore is calculated from the peptide confidence that has not yet been ‘used’ by higher scoring proteins in the experiments. ^b^: The percentage of matching amino acids from the identified peptides. ^c^: The number of distinct peptides with at least 95% confidence in the experiments. ^d^: The ratio represents 115/114 ratio. 115 and 114 were used to label exosomal proteins from J20 mice and wild type mice, respectively. iTRAQ: isobaric tag for relative and absolute quantitation. U-exo: urine exosomes.

**Table 2 ijms-24-00672-t002:** The different proteins we identified in N-exo between WT and J20 groups.

Unused ProtScore ^a^	% Coverage ^b^	UniProt Accession	Gene Symbol	Name	Peptides (95%) ^c^	115:114
6.38	39.5	Q6UGQ3	Scgb2b2	Secretoglobin family 2B member 2	3	3.6573
14.09	48.5	Q9D3H2	Obp1a	Odorant-binding protein 1a	11	2.9136
7.74	8.7	Q8BZH1	Tgm4	Protein-glutamine gamma-glutamyltransferase 4	4	2.7917
2.01	4.5	Q9WU42	Ncor2	Nuclear receptor corepressor 2	1	2.7166
4	17	Q7M747	Scgb2b24	Secretoglobin family 2B member 24	2	2.4941
4	15.1	P18761	Ca6	Carbonic anhydrase 6	3	2.4084
4	24	A2AEP0	Obp1b	Odorant-binding protein 1b	4	2.4014
5.42	66.1	P00687	Amy1	Alpha-amylase 1	36	2.3926
4.28	10.3	P97429	Anxa4	Annexin A4	2	2.2109
6.95	86.7	P11591	Mup5	Major urinary protein 5	50	2.2037
6.09	23.7	Q60590	Orm1	Alpha-1-acid glycoprotein 1	3	2.2033
4.02	34	P01837	Igkc	Ig kappa chain C region	2	2.1829
2.25	5.4	P97434-3	Mprip	Isoform 3 of Myosin phosphatase Rho-interacting protein	1	2.1059
2.28	10.8	Q61147	Cp	Ceruloplasmin	1	2.0868
8.05	18.9	Q07456	Ambp	Protein AMBP	4	2.084
12.89	44.4	O09114	Ptgds	Prostaglandin-H2 D-isomerase	7	1.9728
45.67	44.7	P07724	Alb	Serum albumin	23	1.8624
105.92	51.3	P28825	Mep1a	Meprin A subunit alpha	96	1.8616
2.27	15.2	Q9JI02	Scgb2b20	Secretoglobin family 2B member 20	1	1.8537
4	7.8	Q01279	Egfr	Epidermal growth factor receptor	2	1.852
8.13	35.7	P15501	Sbp	Prostatic spermine-binding protein	5	1.8465
113.02	84.5	P00688	Amy2	Pancreatic alpha-amylase	97	1.8379
3.07	5.4	Q07797	Lgals3bp	Galectin-3-binding protein	1	1.7984
20.01	32.1	P07759	Serpina3k	Serine protease inhibitor A3K	14	1.7874
10.57	81.8	Q5FW60	Mup20	Major urinary protein 20	104	1.7659
3.75	18.1	P46412	Gpx3	Glutathione peroxidase 3	2	1.7549
138.4	87.5	P04939	Mup3	Major urinary protein 3	218	1.7381
46.37	65.6	Q00897	Serpina1d	Alpha-1-antitrypsin 1-4	37	1.7123
14.42	37.7	Q8C6C9	Leg1	Protein LEG1 homolog	10	1.7123
2.07	8.1	Q9CZ52-2	Antxr1	Isoform 2 of Anthrax toxin receptor 1	1	1.7001
2.01	12	O35298	Aoah	Acyloxyacyl hydrolase	1	1.697
24.53	52.5	P15947	Klk1	Kallikrein-1	23	1.6927
10.38	33	P13634	Ca1	Carbonic anhydrase 1	6	1.6857
2.57	16.6	P06869	Plau	Urokinase-type plasminogen activator	1	1.6756
5.67	58.4	Q00898	Serpina1e	Alpha-1-antitrypsin 1-5	29	1.661
11.55	64.4	P07758	Serpina1a	Alpha-1-antitrypsin 1-1	36	1.6561
5.13	11.4	P01898	H2-Q10	H-2 class I histocompatibility antigen, Q10 alpha chain	2	1.6519
19.02	29	Q9R1E6	Enpp2	Ectonucleotide pyrophosphatase/phosphodiesterase family member 2	11	1.6432
4.1	9.1	Q61391	Mme	Neprilysin	2	1.629
4.01	12.6	Q99LJ1	Fuca1	Tissue alpha-L-fucosidase	2	1.6238
6.05	29.2	P10605	Ctsb	Cathepsin B	3	1.6108
2.02	7.9	P04925	Prnp	Major prion protein	1	1.5938
4.17	10.4	Q571E4	Galns	N-acetylgalactosamine-6-sulfatase	2	1.5885
18.01	42.1	P03953	Cfd	Complement factor D	12	1.5799
6.15	28.8	P02816	Pip	Prolactin-inducible protein homolog	3	1.5788
10.89	11.5	P97449	Anpep	Aminopeptidase N	6	1.562
19.03	20.7	P23780	Glb1	Beta-galactosidase	10	1.5572
3.52	11.9	Q61592	Gas6	Growth arrest-specific protein 6	2	1.5507
6	10.6	P70269	Ctse	Cathepsin E	4	1.5426
22.14	20.2	P09470	Ace	Angiotensin-converting enzyme	11	1.5142
4.02	16.5	Q9DBV4	Mxra8	Matrix remodeling-associated protein 8	2	1.5131
2.02	7.9	P23953	Ces1c	Carboxylesterase 1C	1	1.5105
30.67	50	P49183	Dnase1	Deoxyribonuclease-1	17	1.4851
4	9	P20060	Hexb	Beta-hexosaminidase subunit beta	2	1.4841
4.29	86.7	P11588	Mup1	Major urinary protein 1	331	1.4812
6.03	13	Q61398	Pcolce	Procollagen C-endopeptidase enhancer 1	3	1.469
3.7	16.4	Q60963	Pla2g7	Platelet-activating factor acetylhydrolase	2	1.4604
8.6	13.7	P16675	Ctsa	Lysosomal protective protein	4	1.4516
4.2	15.2	Q9Z0K8	Vnn1	Pantetheinase	4	1.447
4	15.9	P63101	Ywhaz	14-3-3 protein zeta/delta	2	1.4451
7.78	17.1	Q6GU68	Islr	Immunoglobulin superfamily containing leucine-rich repeat protein	4	1.4442
4.06	9.4	P27046	Man2a1	Alpha-mannosidase 2	2	1.4354
16	21.3	Q61847	Mep1b	Meprin A subunit beta	9	1.4333
5.23	16.5	Q9WV54	Asah1	Acid ceramidase	2	1.4315
4.69	13.1	C0HKG6	Rnaset2b	Ribonuclease T2-B	3	1.41
4.05	5.7	O54782	Man2b2	Epididymis-specific alpha-mannosidase	2	1.4055
4	22.5	P19 639	Gstm3	Glutathione S-transferase Mu 3	2	1.3973
5.3	6.1	Q8R0I0	Ace2	Angiotensin-converting enzyme 2	3	1.3865
15.4	30.7	P63260	Actg1	Actin, cytoplasmic 2	10	1.373
2.12	10.4	P24638	Acp2	Lysosomal acid phosphatase	1	1.3441
9.59	5.7	Q5SSE9	Abca13	ATP-binding cassette sub-family A member 13	4	1.3385
5.7	35.9	P62806	Hist1h4a	Histone H4	3	1.3277
5.07	12.3	Q9QWR8	Naga	Alpha-N-acetylgalactosaminidase	2	1.3186
17.26	16.5	O09159	Man2b1	Lysosomal alpha-mannosidase	8	1.3091
2.03	6.5	Q9WUU7	Ctsz	Cathepsin Z	1	1.2977
3.94	14.7	O35082	Kl	Klotho	1	1.2901
2.02	8.1	Q9ESG4	Tmem27	Collectrin	1	1.2855
12.08	29.1	Q9JIL4	Pdzk1	Na(+)/H(+) exchange regulatory cofactor NHE-RF3	6	1.2789
14.34	17.7	Q91WV7	Slc3a1	Neutral and basic amino acid transport protein rBAT	7	1.2712
4	11.7	Q9CYN9	Atp6ap2	Renin receptor	2	1.2534
10.16	43.2	Q9CQ60	Pgls	6-phosphogluconolactonase	5	1.2505
6	12.9	P49935	Ctsh	Pro-cathepsin H	3	1.2393
2.08	7.7	Q8K209	Adgrg1	Adhesion G-protein coupled receptor G1	1	1.2388
4	12.4	Q03404	Tff2	Trefoil factor 2	2	1.2351
4.06	19.2	O89051	Itm2b	Integral membrane protein 2B	2	1.2343
8.13	26.7	P08226	Apoe	Apolipoprotein E	5	1.2285
5.7	10.4	Q9CZT5	Vasn	Vasorin	3	1.2261
16.33	10.9	Q61838	Pzp	Pregnancy zone protein	8	1.226
63.56	61.2	Q06890	Clu	Clusterin	50	1.2243
23.36	25.4	P70699	Gaa	Lysosomal alpha-glucosidase	16	1.2238
12.12	22.1	O89023	Tpp1	Tripeptidyl-peptidase 1	6	1.2174
3.3	16.1	P63082	Atp6v0c	V-type proton ATPase 16 kDa proteolipid subunit	3	1.2157

Table 2 shows the different proteins between J20 and WT meeting the following criteria: (1) FDR < 5%. (2) The number of distinct peptides with at least 95% confidence ≥1. (3) Fold change 1.2. ^a^: Unused ProtScore is a score of protein confidence, and reflects the amount of total and unique peptide evidence related to an identified protein. Unused ProtScore is calculated from the peptide confidence that has not yet been ‘used’ by higher scoring proteins in the experiments. ^b^: The percentage of matching amino acids from the identified peptides. ^c^: The number of distinct peptides with at least 95% confidence in the experiments. 115 was used for labelling exosomal proteins of J20 mouse, and 114 was used for labelling exosomal proteins of WT mouse. iTRAQ: isobaric tag for relative and absolute quantitation. N-exo: neuron-derived exosomes.

**Table 3 ijms-24-00672-t003:** Neuron markers in U-exo and N-exo.

Gene Symbol	UniProt Accession	Name	U-exo	N-exo
Zfp292	Q9Z2U2	Zinc finger protein 292	√	√
Spen	Q62504-3	Isoform 3 of Msx2-interacting protein	×	√
Smg1	Q8BKX6	Serine/threonine-protein kinase SMG1	√	√
Ryr2	E9Q401	Ryanodine receptor 2	√	√
Ptgds	O09114	Prostaglandin-H2 D-isomerase	√	√
Myt1l	P97500	Myelin transcription factor 1-like protein	×	√
Myo9a	Q8C170	Unconventional myosin-IXa	√	√
Morc2a	Q69ZX6	MORC family CW-type zinc finger protein 2A	×	√
Kif21b	Q9QXL1	Kinesin-like protein KIF21B	√	√
Kdm2b	Q6P1G2	Lysine-specific demethylase 2B	×	√
Kcna4	Q61423	Potassium voltage-gated channel subfamily A member 4	×	√
Igkc	P01837	Ig kappa chain C region	×	√
Epc2	Q8C0I4	Enhancer of polycomb homolog 2	×	√
Cacna1b	O55017	Voltage-dependent N-type calcium channel subunit alpha-1B	×	√
Atp1b1	P14094	Sodium/potassium-transporting ATPase subunit beta-1	√	√

After checking the detected proteins in U-exo and N-exo, the neuron markers are shown in Table 3. The neuron marker reference online database was CellMarker 2.0 (http://117.50.127.228/CellMarker/ accessed on 10 September 2022). √ stands for presence, and × stands for absence. U-exo: urine exosomes. N-exo: neuron-derived exosomes.

**Table 4 ijms-24-00672-t004:** Summary of proteins enriched in five BP terms with high enrichment in both U-exo and N-exo.

BP Terms	U-exo		N-exo	
	Gene Symbol	Fold Change	Gene Symbol	Fold Change
Sphingolipid catabolic process	Asah1	1.3824	Asah1	1.4315
	Galc	1.1177	Galc	0.9068
	Hexb	1.1909	Hexb	1.4841
	Smpdl3a	1.3852	Smpdl3a	1.1356
	Smpd1	1.3335	Naga	1.3186
			Enpp2	1.6432
Ceramide catabolic process	Asah1	1.3824	Asah1	1.4315
	Galc	1.1177	Galc	0.9068
	Hexb	1.1909	Hexb	1.4841
			Naga	1.3186
Membrane lipid catabolic process	Asah1	1.3824	Asah1	1.4315
	Galc	1.1177	Galc	0.9068
	Hexb	1.1909	Hexb	1.4841
	Smpdl3a	1.3852	Smpdl3a	1.1356
	Smpd1	1.3335	Naga	1.3186
			Enpp2	1.6432
Aβ clearance	Apoe	1.4288	Apoe	1.2285
	Clu	1.3055	Clu	1.2243
	Lrp2	1.1960	Lrp2	1.16
	Mme	1.6804	Mme	1.629
Aβ metabolic process	Ace	1.5564	Ace	1.5142
	Apoe	1.4288	Apoe	1.2285
	Clu	1.3055	Clu	1.2243
	Mme	1.6804	Mme	1.629
			Prnp	1.5938

BP: biological process. U-exo: urine exosomes. N-exo: neuron-derived exosomes.

## Data Availability

All data in this study are shown in the report.

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
