# Peer review of "iTRAQ-Based Proteomic Analysis of APP Transgenic Mouse Urine Exosomes"

_ijms, 2022, doi:10.3390/ijms24010672_

Round 1

Reviewer 1 Report

The article by Zhou et al. aimed to specifically find different proteins under early AD pathology in the urine exosome (U-Exo) of the AD mouse model (J20). The U-Exo was isolated from 3-month-old J20 and wild-type mice, then neuron-derived exosome (N-exo) was separated by immunoprecipitation, and finally, iTRAQ-based MALDI TOF MS/MS was used for proteomic analysis. Proteomic data were analyzed using bioinformatics tools and identified 79 (61 upregulated proteins) and 117 (92 upregulated proteins) proteins specifically expressed in U-Exo and N-Exo, respectively. Additionally, GO enrichment and protein-protein interaction analyses were performed on the proteins. Authors finally highlighted acid ceramidase (Asah1), a key protein in lipid metabolism, and 4 other proteins specific for Aβ pathology, including Clusterin, apolipoprotein E, neprilysin (Mme), and angiotensin, and expected it to be candidate markers for the diagnosis of AD. Although the manuscript covers important and current topics, including AD and exosomes, and shows some promise to have a non-invasive screening in other bodily fluids, such as urine e, to diagnose AD, it lacks a number of things as below-

1.       The manuscript is based on pooled samples, with no biological replicates and only a few technical replicates- and thus, no statistical tests were performed in this work, and even authors point it out in the manuscript. Moreover, this work is solely on a mouse model and not on human samples- thus, it lacks the significance of outlining any candidate AD biomarker

2.      Authors also show a method to extract N-exo from urine- and perform some quality control to make sure N-Exo was extracted- but most of the morphological and other parameters, including size between N-exo and U-Exo, were the same- which makes it a bit hard to believe that pure N-exo were enriched.

3.      Although this method could be useful for the community, I think it will be important to repeat this work at least on one more set of pooled samples- to make at least two biological replicates (although usually at least 3 are required for any statistical test)- to reproduce the findings- as the statistical test would require at least 3 replicates.

4.      Some of the parameters for proteomics used by authors also raise concerns- they use peptides of more than 1 and fold change (which is actually not a statistical fold change- and is misleading) of 1.2.

5.      The number of proteins identified is too low- why is that- is it normal to see these fewer proteins for urine exosome work? References are needed.

6.      Page 9- 2.4- it says figure 1b- which I think is wrong?

7.      Were identified or differentially expressed proteins used for the GO and protein interaction and other bioinformatic work

8.      The western blot was only performed for Clusterin and not for other proteins, and this seems like it was also done just one time- no replicates.

9.      The author uses low pvalue, enrichment scores, etc., for all the analyses performed- were these obtained from software? How are these calculated? The author has commented that they couldn’t perform statistical tests- thus, the use of these pvalues makes it misleading.

10.  Use of terminology- upregulated and downregulated is not apt for protein abundance obtained from proteomics; thus, this should be changed to increased or decreased abundance.

11.  Only male mice were used. And only 3 months age- for comparison with AD- wouldn’t aged mice would be appropriate?

12.  The author says urine was collected for 1month- how much, how many intervals?

13.  Page 20, 4.2- the author said they used 3 pooled samples for Uexo and 9 for Nexo- why was this discrepancy? Would this cause any changes between them?

14.  For trypsin-what was the substrate enzyme ratio used?

15. Discussion is mainly a repeat of results, more discussion about findings is needed.

Overall, I think before this manuscript could be accepted- authors need at least to redo this on one more set of pooled samples- to show the results are accurate and use some more stringent parameters. In this form, I don’t think the manuscript is publishable. But, the work done is good; thus, I am not rejecting the manuscript but putting it as a major review- so that authors could improve the manuscript based on my comments and resubmit.

Author Response

Review 1

Comments and Suggestions for Authors

The article by Zhou et al. aimed to specifically find different proteins under early AD pathology in the urine exosome (U-Exo) of the AD mouse model (J20). The U-Exo was isolated from 3-month-old J20 and wild-type mice, then neuron-derived exosome (N-exo) was separated by immunoprecipitation, and finally, iTRAQ-based MALDI TOF MS/MS was used for proteomic analysis. Proteomic data were analyzed using bioinformatics tools and identified 79 (61 upregulated proteins) and 117 (92 upregulated proteins) proteins specifically expressed in U-Exo and N-Exo, respectively. Additionally, GO enrichment and protein-protein interaction analyses were performed on the proteins. Authors finally highlighted acid ceramidase (Asah1), a key protein in lipid metabolism, and 4 other proteins specific for Aβ pathology, including Clusterin, apolipoprotein E, neprilysin (Mme), and angiotensin, and expected it to be candidate markers for the diagnosis of AD. Although the manuscript covers important and current topics, including AD and exosomes, and shows some promise to have a non-invasive screening in other bodily fluids, such as urine e, to diagnose AD, it lacks a number of things as below.

  1. The manuscript is based on pooled samples, with no biological replicates and only a few technical replicates- and thus, no statistical tests were performed in this work, and even authors point it out in the manuscript. Moreover, this work is solely on a mouse model and not on human samples- thus, it lacks the significance of outlining any candidate AD biomarker

Re: Thank you so much for your comments.

              Generally speaking, there are many types of iTRAQ experimental designs, among which, in many studies, each sample is used for one iTRAQ experiment and repeat several times (PMID: 24435478, PMID: 22552374, PMID: 33569037). However, pooled samples are also widely used in iTRAQ, including several different pooled samples was prepared in the same group for biological replicates (PMID: 29626725, PMID: 33184749, PMID: 34900729, PMID: 32736138), and all samples in a group are pooled for technical replicates (PMID: 26425554, PMID: 29068691), and all samples in a group are pooled for one-time iTRAQ without replicates (PMID: 29355489, PMID: 28009970, PMID: 34571828, PMID: 17853512, PMID: 28800743).

Sample pooling strategy has been used widely to reduce the effect of biological variation (PMID: 28800743, PMID: 24448401, PMID: 17675576, PMID: 26966714, PMID: 32736138), so we use pooled samples in this study. For U-exo, we used a total of 9 mice in one group, and every 3-mouse urine was pooled into one sample, and there were 3 pooled samples in total, all them were used for iTRAQ and MALDI-TOF MS/MS, so I think we performed biological replicates (PMID: 29626725, PMID: 33184749, PMID: 34900729, PMID: 32736138). The N-exo is extracted from U-exo by immunoprecipitation, so the volume is less, we used a mixture of urine from 9-mouse sample, and we did not repeat it (PMID: 29355489, PMID: 28009970, PMID: 34571828, PMID: 17853512, PMID: 28800743).

We agree with you that the important proteins we identified in this study are not suitable as AD biomarkers and have been changed in the manuscript accordingly. The corresponding discussion about AD mouse model was also added (page 20, line 420-433).

  1. Authors also show a method to extract N-exo from urine- and perform some quality control to make sure N-Exo was extracted- but most of the morphological and other parameters, including size between N-exo and U-Exo, were the same- which makes it a bit hard to believe that pure N-exo were enriched.

Re: We are very grateful for your valuable comments.

We established this method according to previous study (PMID: 28588440) where Maja Mustapic et al. demonstrated that the method is suitable for isolating neuron-derived exosomes from the peripheral circulation. Their results showed that compared with total plasma exosomes, L1CAM+ plasma exosomes contained a series of higher levels of neuronal proteins, such as neuronal enolase (NSE), neural cell adhesion molecule (NCAM), and neurofilaments-light (NFL), and pro-brain derived neurotrophic factor (pro-BDNF) (PMID: 28588440), suggesting that neuron-derived exosomes were successfully enriched by this method. Up to now, this method has been widely used to isolate neuron-derived exosomes (PMID: 28692534, PMID: 32002168, PMID: 25130657, PMID: 32273329, PMID: 30605353, PMID: 35159246,PMID: 33418508).

In this study, transmission electron microscopy and western blot were used to verify the success to extraction of exosomes, because exosomes need to meet the following criteria: 1. Exosomes have a typical spherical bilayer membrane structure. 2. The diameter of exosomes is in the range of 30-150nm. 3. Expression of exosomal markers, including ALIX, Tsg101, CD63 and CD9 (PMID: 31311206).

Furthermore, in order to verify the efficacy of the N-exo isolation method, we compared the proteins which can be expressed in brain between U-exo and N-exo by web tool GenAtlas (http://genatlas.medecine.univ-paris5.fr/). Our results showed that 30 % U-exo proteins (24 proteins) can be expressed in brain while 35% N-exo proteins (41 proteins) can be expressed in brain (Tables S3 and S4, Figure 4). Therefore, our results confirmed that N -exo contains more proteins that can be expressed in the brain, indicating that neuron-derived exosomes are well enriched from U-exo through L1CAM antibody. The content was added to the corresponding Results (Page 9, line 192-197) and Discussion section (Page 21-22, line 501-507).

  1. Although this method could be useful for the community, I think it will be important to repeat this work at least on one more set of pooled samples- to make at least two biological replicates (although usually at least 3 are required for any statistical test)- to reproduce the findings- as the statistical test would require at least 3 replicates.

Re: Thank you very much for your precious suggestions.

Generally speaking, there are many types of iTRAQ experimental designs, among which, in many studies, each sample is used for one iTRAQ experiment and repeat several times (PMID: 24435478, PMID: 22552374, PMID: 33569037). However, pooled samples are also widely used in iTRAQ, including several different pooled samples was prepared in the same group for biological replicates (PMID: 29626725, PMID: 33184749, PMID: 34900729, PMID: 32736138), and all samples in a group are pooled for technical replicates (PMID: 26425554, PMID: 29068691), and all samples in a group are pooled for one-time iTRAQ without replicates (PMID: 29355489, PMID: 28009970, PMID: 34571828, PMID: 17853512, PMID: 28800743).

Sample pooling strategy has been used widely to reduce the effect of biological variation (PMID: 28800743, PMID: 24448401, PMID: 17675576, PMID: 26966714, PMID: 32736138), so we use pooled samples in this study. For U-exo, we used a total of 9 mice in one group, and every 3-mouse urine was pooled into one sample, and there were 3 pooled samples in total, all them were used for iTRAQ and MALDI-TOF MS/MS, so I think we performed biological replicates (PMID: 29626725, PMID: 33184749, PMID: 34900729, PMID: 32736138). The N-exo is extracted from U-exo by immunoprecipitation, so the volume is less, we used a mixture of urine from 9-mouse sample, and we did not repeat it (PMID: 29355489, PMID: 28009970, PMID: 34571828, PMID: 17853512, PMID: 28800743).

  1. Some of the parameters for proteomics used by authors also raise concerns- they use peptides of more than 1 and fold change (which is actually not a statistical fold change- and is misleading) of 1.2.

Re: Thank you very much for your constructive comments.

              To minimize the false positive identification of proteins, we used a cutoff of the number of distinct peptides with at least 95% confidence ≥ 1 as the qualification criteria. In mass spectrometry, the number of peptides is often used as an important parameter for protein qualification criteria. (PMID: 23802875, PMID: 22552374, PMID: 33250918, PMID: 33184749).

ITRAQ is used to quantify the relative abundance of peptides and proteins across the samples. So, in this experiment, the data we obtained is only the ratio of the relative abundance of proteins between the J20 and WT groups. 1.2-fold change is often used to screen different proteins (PMID: 33519723, PMID: 31115557, PMID: 32705285, PMID: 33509101, PMID:30002204, PMID: 35571066, PMID: 33184749)

  1. The number of proteins identified is too low- why is that- is it normal to see these fewer proteins for urine exosome work? References are needed.

Re: We are very grateful for your meaningful question and advice. And I’m sorry for confusing your reading.

In this article, not all proteins detected by the MALDI TOF MS/MS were selected as identified proteins. A total of 659 proteins with intensity values were detected in U-exo (Table S1), and a total of 481 proteins with intensity values were detected in N-exo (Table S2). In order to minimize the false positive identification of proteins, we only selected the identified proteins based on criteria:  FDR<5% (PMID: 26709396, PMID: 26078484), the number of distinct peptides with at least 95% confidence ≥ 1, and the protein was detected with ratio (PMID: 26078484, PMID: 24435478). After selection, 79 U-exo proteins and 117 N-exo proteins were screened as identified proteins (Tables S3, S4). Now, we have summarized all the proteins detected by MALDI TOF MS/MS with signal intensities in Table S1 and Table S2, and the content is added in the Results section (Page 4-5, line 138-150).

  1. Page 9- 2.4- it says figure 1b- which I think is wrong?

Re:Thanks for your comment. After checking, we found this is a mistake. Now, we have corrected it.

  1. Were identified or differentially expressed proteins used for the GO and protein interaction and other bioinformatic work

Re: We appreciate your comment and we are sorry for the unclear description.

In this paper, both the common different proteins (Figure 5) and the identified proteins (Figure 6) were used for GO analysis. And the identified proteins were also used for protein interaction analysis (Figure 8).

  1. The western blot was only performed for Clusterin and not for other proteins, and this seems like it was also done just one time- no replicates.

Re: Thank you very much for your important comment.

For western blot, the N-exo is extracted from U-exo by immunoprecipitation, so the volume is less, and in order to obtain more neuron-derived exosome proteins, we used a pooled sample from 9 mice, so no replication was performed. And our purpose is to focus on the clusterin existence to verify the authenticity of the protein identified by iTRAQ, rather than statistical analysis. Since it is difficult to obtain enough N-exo proteins for the verification of more proteins, we selected the most important proteins for western blot. Among the important proteins (Asah1, clusterin, ApoE, neprilysin, and ACE), clusterin has important features: 1. clusterin had markedly higher peptide coverage with 49 and 50 peptides in U-exo and N-exo, respectively (Figure 9A), suggesting that clusterin has the highest confidence and abundance. 2.clusterin was enriched in BP terms performed by the common different proteins (Table S6). 3.clusterin was enriched in the overlapped BP terms Aβ clearance and Aβ metabolic process (Table 3). 4. clusterin was a member of MCODE complex with highest score (Figure 8). 5. It was previously reported that clusterin plays an important role in AD and has the potential to become a marker of AD (PMID: 26488311, PMID: 30770953). Considering the limitation of N-exo, we finally chose clusterin for WB. An explanation of the selection of clusterin for western blot has been added in the Results section (Page 18, line 375-385).

We are sorry that due to the limitation of N-exo samples, we cannot do more western blots. But we will use human samples to verify the total important proteins we found in this study, and we are currently collecting human urine.

  1. The author uses low pvalue, enrichment scores, etc., for all the analyses performed- were these obtained from software? How are these calculated? The author has commented that they couldn’t perform statistical tests- thus, the use of these pvalues makes it misleading.

Re: We are very thankful your significant comments.

In bioinformatics analysis, both p value and enrichment score were calculated by the software Metascape (https://metascape.org/), which was calculated by comparing the identified proteins with the background proteins in the database. The value reflects the enrichment degree and probability of the enriched terms, rather than the evaluation of a certain protein. Pathway enrichment is measured by p value and enrichment factor. The p values are calculated based on the accumulative hypergeometric distribution. Enrichment factor is calculated by dividing hit ratio by background ratio. Hit ratio represents the ratio of the number of genes in enriched pathway to the total number of genes used for analysis. The background ratio represents the ratio of the number of annotated genes in the enriched pathway to the number of all background genes. Enrichment factor indicates how many fold times more given pathway members are found in our gene list compared to what would have been expected by chance. The value range of p value is [0,1]. The closer it is to zero, the more significant the enrichment is. Pathways with p < 0.05 are defined as pathways that are significantly enriched. The larger the enrichment factor, the greater the degree of enrichment. Interpretations of p-values and enrichment factors are shown in the Methods section (Page 27, line 775-785).

  1. Use of terminology- upregulated and downregulated is not apt for protein abundance obtained from proteomics; thus, this should be changed to increased or decreased abundance.

Re: We greatly appreciate your valuable advice. We have changed it.

  1. Only male mice were used. And only 3 months age- for comparison with AD- wouldn’t aged mice would be appropriate?

Re: We sincerely appreciate for your important advice.

Female mice have menstrual cycle, so it is difficult to ensure that each female mouse is in the same state. To reduce biological differences and ensure the comparability of results, we selected male mice with a more stable state.

Three-month-old mice were chosen because we aimed to screen important proteins in the early stages of AD pathology. For diagnosis and intervention, early diagnosis is more conducive to the prevention and treatment of AD. Of course, a time-dependent study would provide more detailed information, which would be helpful to select a disease diagnosis marker in the urine samples. This will be the direction of our future research. Now, we add this content to the limitations of the Discussion section (Page 23, line 607-611).

  1. The author says urine was collected for 1month- how much, how many intervals?

Re:  We really appreciate your precious question. And we are sorry for not describing the process of sample collection in detail.

Mouse urine was collected daily for 1 month from 12 to 16 weeks. The total volume of urine in wild-type mice is about 20 ml and that in J20 mice is about 15 ml. A more detailed procedure for sample collection has been added to the Materials and Methods section of the manuscript (Page 25, line 647-649).

  1. Page 20, 4.2- the author said they used 3 pooled samples for Uexo and 9 for Nexo- why was this discrepancy? Would this cause any changes between them?

Re: We are very grateful for your constructive comments.

For each group of mice, we collected urine from a total of 9 mice in every group. For U-exo, in order to ensure that we can obtain enough urine exosome proteins and carry out biological replication, we choose pooled sample from 3 mice. However, neuron-derived exosomes are isolated from U-exo by immunoprecipitation, so the volume is less, we pooled sample from 9 mice. Mixed samples can reduce biological differences, and U-exo and N-exo were isolated from the same mice, the experimental results are comparable.

  1. For trypsin-what was the substrate enzyme ratio used?

Re: Thank you very much for your question.

When we were preparing the samples for the iTRAQ experiment, the substrate enzyme ratio was 30:1(w/w). This is a detail that we missed when writing, and it has been added in the materials and methods section of the manuscript (Page 26, line 730-732).

  1. Discussion is mainly a repeat of results, more discussion about findings is needed.

Re: We sincerely appreciate your constructive questions. We have revised the Discussion section to improve it.

Overall, I think before this manuscript could be accepted- authors need at least to redo this on one more set of pooled samples- to show the results are accurate and use some more stringent parameters. In this form, I don’t think the manuscript is publishable. But, the work done is good; thus, I am not rejecting the manuscript but putting it as a major review- so that authors could improve the manuscript based on my comments and resubmit.

Reviewer 2 Report

Authors have done a decent job to isolate exosomes from urine and brain samples to delineate specific marker proteins in AD models. However there are multiple caveats in the study. The whole study is based on one single isolation and MS data. Validation of MS data is also not very convincing. Among many other these are few concerns that authors should address, before resubmitting the paper.

Abstract is poorly written. Please provide specifically what are the major findings. For example, it is mentioned that neuronal exo was isolated by IP, while there is no mention of method used for U-exo isolation. Also, name specific pathways and genes identified throgh the study. What does ".........U-exo and N-exo mainly mediate the degradation process" mean? What kind of degradation U-exo carries? Please specify.

It is not clear from the description what authors are comparing these proteins with when they say 61 and 92 proteins were upregulated. Please mention clearly What are numerators and denominators in calculating ratios.

Is it true that authors only identified 71 and 117 proteins in total for both preps respectively? What was no. of replicates for this.  Please elaborate more in writing. apart from table legend please include crucial details in your main text (results).

Please provide a comparison of proteins (venn) being present in both the lists (Table 1 and 2), if any. Perform GO enrichment of those to see the most important terms.

Figure 2: For subcellular localization plots, please provide bar plots with p values, enrichment and FDR values. Presentation in for of % does not cover much necessary information for such analysis. For example, there could be overlapping proteins between extracellular and plasma membrane-like terms. How did authors categorize those terms?

Highlight which terms are common for both datasets, u-exo and--exo in GO analysis bar graphs in FIgure 3.

What is rationale behind "2.6 Analysis of exosome proteins enriched in biological processes terms.". I didn't get the context. Does it help the study?

What are Abeta related BP terms? Please avoid writing non-scientific jargons.

Why the comparison among MCODE1 to 4  for u-Exo and n-Exo is done? Are these same proteins? Please follow same color code in A, B and C for different clusters.

Why did authors choose clusterin among all other proteins detected? any specific criteris. Is high spec count or number of peptides obsered is he only criteria? Please verify some other low abundant proteins too to avoid any misinterpretation of the data. ALso state clearly the criteria and rationale (including functional or biological relevance) behind selection of proteins.

Overall writing of the paper is weak. The data could be summarized and presented in a better way. Consolidating figures and providing better description of data, a number of replicates for each study would help addressing some of these issue.

Author Response

Review 2:

Comments and Suggestions for Authors

Authors have done a decent job to isolate exosomes from urine and brain samples to delineate specific marker proteins in AD models. However there are multiple caveats in the study. The whole study is based on one single isolation and MS data. Validation of MS data is also not very convincing. Among many other these are few concerns that authors should address, before resubmitting the paper.

1.Abstract is poorly written. Please provide specifically what are the major findings. For example, it is mentioned that neuronal exo was isolated by IP, while there is no mention of method used for U-exo isolation. Also, name specific pathways and genes identified throgh the study. What does ".........U-exo and N-exo mainly mediate the degradation process" mean? What kind of degradation U-exo carries? Please specify.

Re: Thank you much for your essential comments. We have revised the Abstract section according to your suggestions (Page 1, line 15-28).

2.It is not clear from the description what authors are comparing these proteins with when they say 61 and 92 proteins were upregulated. Please mention clearly What are numerators and denominators in calculating ratios.

Re:  Thank you very much for your valuable comments, and I am also sorry for the troubles you read due to the unclear description.

In this study, 114 was used for labeling wild-type, 115 was used for transgenic-type, the ratio reflected 115/114 (J20/WT). Criteria for screening differential proteins were added to the Results section (Page 4-5, line 138-152).

  1. Is it true that authors only identified 71 and 117 proteins in total for both preps respectively? What was no. of replicates for this. Please elaborate more in writing. apart from table legend please include crucial details in your main text (results).

Re: Thanks a lot for your questions, and we are sorry for the reading obstacles caused by my insufficiently detailed writing.

In this article, not all proteins detected by the MALDI TOF MS/MS were selected as identified proteins. A total of 659 proteins with intensity values were detected in U-exo (Table S1). A total of 481 proteins with intensity values were detected in N-exo (Table S2). In order to minimize the false positive identification of proteins, we only selected the identified proteins based on the criteria: FDR<5% (PMID: 26709396, PMID: 26078484), the number of distinct peptides with at least 95% confidence ≥ 1, and the protein was detected with ratio (PMID: 26078484, PMID: 24435478). After selection, 79 U-exo proteins and 117 N-exo proteins were screened as identified proteins (Tables S3, S4). Now, we have summarized all the proteins detected by MALDI TOF MS/MS with signal intensities in Table S1 and Table S2, and the content is added in the Results section (Page 4-5, line 138-147).

For U-exo, we used a total of 9 mice in every group, and every 3-mouse urine was pooled into one sample, and there were three pooled samples in total, all them were used for iTRAQ and MALDI-TOF MS/MS, so we performed biological replicates 3 times (PMID: 29626725, PMID: 33184749, PMID: 34900729, PMID: 32736138). The N-exo is extracted from U-exo, so the content is less, we used a mixture of urine from 9 mouse sample (PMID: 29355489, PMID: 28009970, PMID: 34571828, PMID: 17853512), and we did not repeat it.

  1. Please provide a comparison of proteins (venn) being present in both the lists (Table 1 and 2), if any. Perform GO enrichment of those to see the most important terms.

Re: We are so thankful for your vital advice.

We performed comparison of proteins being present in both the lists (Table 1 and 2), the comparison Venn diagram was shown as Figure 2, and GO enrichment analysis of the common different proteins was shown in Figure 5, Table S5 and Table S6. The corresponding results section has also been added (Page 5 line 152-155, page 11-12 line 212-246).

  1. Figure 2: For subcellular localization plots, please provide bar plots with p values, enrichment and FDR values. Presentation in for of % does not cover much necessary information for such analysis. For example, there could be overlapping proteins between extracellular and plasma membrane-like terms. How did authors categorize those terms?

Re: We greatly appreciate your suggestions.

We performed subcellular localization prediction using the online tool WoLF PSORT (https://psort.hgc.jp/) (PMID: 17517783). According to species and multifasta format protein sequence, WoLF PSORT will calculate scores for possible localization sites. The larger the value, the more likely the protein is localized at the position.

For example, sp|Q06890|CLUS_MOUSE WoLFPSORT prediction extr: 15, extr_plas: 11, plas: 5, nucl: 4.5, cyto_nucl: 3.5, E.R.: 3, lyso: 3, cyto: 1.5.

The maximum score is 15, and its corresponding subcellular localization is extracellular, so the most likely subcellular localization of mouse clusterin (Q06890) is extracellular. In this study, we only select the subcellular localization corresponding to the largest scores for possible localization sites, and this method has also been reported for subcellular structure prediction (PMID: 26843850, PMID: 34572535, PMID: 35360110, PMID: 31715689, PMID: 32572646, PMID: 33408763, PMID: 34922597). The complete predictions of subcellular localization are placed in the Table S3 and Table S4. A more detailed explanation has been added to the figure legend (Page 10, line 199-205).

  1. Highlight which terms are common for both datasets, u-exo and--exo in GO analysis bar graphs in FIgure 3.

Re: Thanks a lot for your meaningful advice. We have highlighted the terms common for both U-exo and N-exo: in Figure 6, common terms in U-exo and N--exo are shown in red font.

  1. What is rationale behind "2.6 Analysis of exosome proteins enriched in biological processes terms.". I didn't get the context. Does it help the study?

Re: We are grateful for your significant question. The title is not good. We have changed it into ‘2.7 Comparison of exosome proteins participated in the highly enriched BP terms in both U-exo and N-exo’ (Page 16 line 307-308). This analysis was performed to screen for proteins that play an important role in biological processes in the early pathological stages of AD. The corresponding explanations have been placed in the Results section (Page16, line 318-324).

  1. What are Abeta related BP terms? Please avoid writing non-scientific jargons.

Re: Thank you so much for your meaningful advice. We have changed it.

  1. Why the comparison among MCODE1 to 4 for u-Exo and n-Exo is done? Are these same proteins? Please follow same color code in A, B and C for different clusters.

Re: We deeply appreciate for your constructive questions and suggestions.

MCODE 1-4 is protein complexes obtained from protein-protein interaction analysis in U-exo and N-exo. The comparison reveals similarities and differences in the composition of protein complexes. Due to the different protein profiles of U-exo and N-exo, composition of protein complex is not identical. Usually, proteins can hardly function as isolated species in the body, and over 80% of proteins do not function individually but in complexes [PMID: 24693427]. Therefore, protein-protein interaction analysis helps to find interactors of target proteins and reveal protein functions. Protein complexes tend to interact and function together, and proteins contained in protein complexes with the highest enrichment scores in both U-exo and N-exo showed the more important functions in the early stages of AD pathology.

Changes have been made to the figure 8 so that the same colors reflect the same clusters.

  1. Why did authors choose clusterin among all other proteins detected? any specific criteris. Is high spec count or number of peptides obsered is he only criteria? Please verify some other low abundant proteins too to avoid any misinterpretation of the data. ALso state clearly the criteria and rationale (including functional or biological relevance) behind selection of proteins.

Re: Thank you very much for your precious questions and suggestions.

For western blot, the N-exo is extracted from U-exo by immunoprecipitation, so the volume is less. Since it is difficult to obtain enough U-exo proteins for the verification of more proteins, we selected the most important proteins for western blot. Among the important proteins (Asah1, clusterin, ApoE, neprilysin, and ACE), clusterin has important features: 1. clusterin had markedly higher peptide coverage with 49 and 50 peptides in U-exo and N-exo, respectively (Figure 9A), suggesting that clusterin has the highest confidence and abundance. 2.clusterin was enriched in BP terms performed by the common different proteins (Table S6). 3.clusterin was enriched in the overlapped BP terms Aβ clearance and Aβ metabolic process (Table 3). 4. clusterin was a member of MCODE complex with highest score (Figure 8). 5. It was previously reported that clus-terin plays an important role in AD and has the potential to become a marker of AD (PMID: 26488311, PMID: 30770953, PMID:31878951). Considering the limitation of N-exo, we finally chose clusterin for WB. An explanation of the selection of clusterin for western blot has been added in the Results section (Page 18, line 375-385).

Overall writing of the paper is weak. The data could be summarized and presented in a better way. Consolidating figures and providing better description of data, a number of replicates for each study would help addressing some of these issue.

Re: We are very grateful for your constructive comments. And we have revised the manuscript and figure sections of the article.